# Intramolecular autoinhibition regulates the selectivity of PRPF40A tandem WW domains for proline-rich motifs

Santiago Martínez-Lumbreras [1,2] ✉, Lena K. Träger[2], Miriam M. Mulorz[3], Marco Payr [2], Varvara Dikaya[2], Clara Hipp [1,2], Julian König [3] & Michael Sattler [1,2] ✉

PRPF40A plays an important role in the regulation of pre-mRNA splicing by mediating protein-protein interactions in the early steps of spliceosome assembly. By binding to proteins at the 5´ and 3´ splice sites, PRPF40A promotes spliceosome assembly by bridging the recognition of the splices. The PRPF40A WW domains are expected to recognize proline-rich sequences in SF1 and SF3A1 in the early spliceosome complexes E and A, respectively. Here, we combine NMR, SAXS and ITC to determine the structure of the PRPF40A tandem WW domains in solution and characterize the binding specificity and mechanism for proline-rich motifs recognition. Our structure of the PRPF40A WW tandem in complex with a high-affinity SF1 peptide reveals contributions of both WW domains, which also enables tryptophan sandwiching by two proline residues in the ligand. Unexpectedly, a proline-rich motif in the N-terminal region of PRPF40A mediates intramolecular interactions with the WW tandem. Using NMR, ITC, mutational analysis in vitro, and immunoprecipitation experiments in cells, we show that the intramolecular interaction acts as an autoinhibitory filter for proof-reading of high-affinity proline-rich motifs in bona fide PRPF40A binding partners. We propose that similar autoinhibitory mechanisms are present in most WW tandem-containing proteins to enhance binding selectivity and regulation of WW/proline-rich peptide interaction networks.

The major spliceosome is a dynamic machinery composed of several dozens of proteins and five small nuclear ribonucleoprotein (snRNP) particles that catalyze in an ordered manner the splicing reaction of most of the pre-mRNAs[1–3]. The sequential assembly around the intron starts with the recognition of the cis sequence elements that characterize the intron/exon boundaries in the so-called complex E: the 5′ splice site (5′SS) and the 3′ regions of the intron, including the branch point sequence (BPS), the polypyrimidine tract (PPT) and the 3′ splice site (3′SS). Initially, the U1 snRNP particle recognizes the 5′SS by base paring of its U1 snRNA[4,5]. The recognition of the 3′SS involves cooperative binding of splicing factor 1 (SF1) and the heterodimeric U2 Auxiliary Factors U2AF2 and U2AF1 to the BPS, the PPT, and the 3′ SS, respectively[6–10]; stabilized by UHM/ULM protein-protein interactions[11–14]. In the next step, complex A, the 17S U2 snRNP particle, replaces SF1 and recognizes the BPS by base paring with the U2 snRNA[15–19]. Subsequent particle remodeling through complexes B

[1]Institute of Structural Biology, Molecular Targets and Therapeutics Center, Helmholtz Munich, Ingolstädter Landstrasse 1, 85764 Neuherberg, Germany. [2]TUM School of Natural Sciences, Department of Bioscience and Bavarian NMR Center, Technical University of Munich, Lichtenbergstrasse 4, 85748 Garching, Germany. [3]Institute of Molecular Biology (IMB) gGmbH, Ackermannweg 4, 55128 Mainz, Germany. ✉e-mail: santiago.martinez@tum.de; michael.sattler@helmholtz-munich.de

and C is necessary to perform the two transesterification reactions and to obtain at the end the intron sequence spliced out (intron lariat) and the mature mRNA[20].

There are several interactions during the early stages of spliceosome assembly, which bridge the 5′ and 3′SS, both across the intervening intron and exon. For example, the stem loop 4 of the U1 snRNA (5′SS) is recognized by the UBL-like domain of SF3A1 (U2 snRNP component in 3′SS)[21,22]; the UAP56 (RNA helicase) binds to the U1 snRNA stem loop 3 enhancing complex A formation[23]; the SR proteins are thought to mediate U1 and U2 snRNPs bridging through their serine-arginine motifs[24–27]; and recently, FUBP1 was identified as a core splicing factor that binds upstream of the BPS but also mediates interactions with the U1 snRNP[28]. The splicing factor PRPF40A was proposed to mediate a bridging connection already in complex E (prior binding of U2 snRNP) by the simultaneous recognition of U1 snRNP at the 5′ SS and SF1 at the BPS[29,30]. In fact, in budding yeast, this cross-intron connection has been clearly demonstrated by the isolation of the complex E and further characterization by mass spectrometry and cryo-electron microscopy (cryo-EM), showing the involvement of Prp40 (the yeast PRPF40A homolog) in establishing a connection between the 3′and 5′SS by the interaction with Luc7 and Snu71 (from the U1 snRNP) and with Msl5 (yeast SF1)[31,32].

PRPF40A is a multidomain protein containing two WW domains and six FF domains in its N- and C-terminal regions, respectively. PRPF40A has been proposed to interact with U1 snRNP components through the C-terminal FF domains in a similar fashion as the yeast counterpart[33]. In addition, the FF domains have been shown to recognize the phosphorylated RNA polymerase II CTD (C-terminal domain), suggesting a role in co-transcriptional splicing regulation[34]. On the other hand, the WW domains of PRPF40A interact with proline-rich proteins, like early splicing factors SF1 or SF3A1 [35], but also with other non-splicing related proteins such as WASP, formin or huntingtin[36–38]. PRPF40A copurifies with early spliceosomal complexes (E and A) and has been classified as a U2 snRNP-associated protein[39]. Its human paralog, PRPF40B, also interacts with SF1 through the WW tandem domains[40] and has been demonstrated to act as an alternative splicing regulator[41]. Finally, the PRPF40A homolog in worms has been recently shown to modulate the splicing of microexons through cross-intron bridging[42]. Therefore, there are multiple links of PRPF40A/B proteins with early splicing events. However, the underlying molecular and structural mechanisms, and functional roles in metazoans remain to be clarified. Notably, PRPF40A has been considered an oncogene and is overexpressed in different cancers[43–46].

The WW domain is a small fold composed of three β-strands arranged in an antiparallel β-sheet, that has been extensively characterized in the past[47–49]. WW domains are able to interact with a great variety of peptides, mainly proline-rich sequences[50]. Its amino acid composition is characterized by the presence of two tryptophan residues, one involved in the formation of small hydrophobic core, and the other being solvent-exposed at the C-terminal end. This tryptophan residue and other aromatic positions conserved on the surface of the β-sheet are involved in establishing interactions, converting this surface into the binding platform for peptide recognition[51,52]. Several studies have analyzed the binding selectivity of WW domains, classifying them in different groups depending on the type of peptide motif recognized[53]; in most cases, WW domains show some promiscuity to bind proline-rich sequences, and recently it has been proposed that this ambiguity can be reduced by the combination of two WW domains in tandem[54,55]. For the PRPF40A WW domains, only the binding of the first one has been extensively characterized, and a short proline-rich motif (PPLP) was identified with an affinity in the high micromolar range[56]. For the second WW domain, a clear interacting motif has not been found[50]. Few studies have explored the WW tandem domains of PRPF40A. It was proposed that the PRPF40 WW tandem forms a compact arrangement, and can bind to proline-rich peptides derived

from huntingtin protein in the medium micromolar range, with about 10-fold increased affinity compared to the individual domains[38]. However, the structural basis for these interactions is unknown. Moreover, it is unclear how the two WW domains of PRPF40A cooperate to recognize proline-rich peptides and what specificity these WW domains show for different proline-rich motifs. Notably, proline-rich motifs recognized by WW domains are often found in large intrinsically disordered regions, where specific high-affinity motifs might exist, but also binding avidity may enhance affinities as seen for other proline-rich binding motifs such as the OCRE domain[57].

Here, we present the structural characterization of the N-terminal region of PRPF40A and its interaction with SF1. Using nuclear magnetic resonance (NMR) in combination with small angle X-ray scattering (SAXS) and isothermal titration calorimetry (ITC), we demonstrate that the WW domains are independent modules, in contrast to a previously reported truncated structure. Our NMR structure shows how both WW domains cooperate to specifically recognize a unique peptide motif within the long and intrinsically disordered C-terminal region of SF1 with 10 times higher affinity compared to any other previously described interaction. Surprisingly, we identified a conserved motif in the N-terminal region of PRPF40A that mediates an intramolecular interaction and thereby proofreads the binding of proline-rich motifs to enhance the selectivity of the WW domains. Our data reveal a structural mechanism for a key protein-protein interaction during early spliceosome assembly. Based on cellular immunoprecipitation experiments and proteomics analysis, we propose that proofreading of proline-rich motif interactions by autoinhibitory intramolecular interactions may represent a general and widespread mechanism used by WW-containing proteins.

## Results
### The WW domains of PRPF40A are folded independently and exhibit significant domain mobility
PRPF40A sequence conservation (Supplementary Fig. 1) and our NMR data acquired for different WW tandem constructs (Supplementary Fig. 2) show that the folded region includes a helical C-terminal extension. Moreover, comparison of NMR spectra of individual WW domains with the tandem (Supplementary Fig. 3) and [15]N NMR relaxation data (Supplementary Fig. 4) indicates the absence of strong interdomain contacts and a certain conformational flexibility of the two domains in solution. This indicates that a previously reported structure of the PRPF40A WW domain tandem[38] is done on a truncated construct. Based on these data, we identified PRPF40A residues 141–236 (hereafter named WW12) as the complete WW tandem domain region, and determined the solution structure of this extended WW tandem (Fig. 1A). Our structure shows two canonical WW domains, comprised of the characteristic three-pleated antiparallel β-sheet and separated by a helical region that interacts with a C-terminal α-helix (Fig. 1A, structural statistics in Supplementary Table 1). The NMR ensemble indicates an overall extended arrangement of the two WW domains with notable structural variations (Supplementary Fig. 5). This is consistent with the lack of interdomain NOEs, and the observation of only a few NOEs connecting the second WW domain with the two α-helices.

To assess the relative domain orientation and validate the structure, several experiments were carried out. Analysis of residual dipolar couplings (RDC) shows different alignment tensors for the individual WW domains in the tandem construct (Supplementary Fig. 6); indicative of the presence of some interdomain mobility[58]. Paramagnetic relaxation enhancement (PRE) data, acquired after the introduction of a paramagnetic label (IPSL) at several positions along the WW12 tandem (see Methods)[59], agree with an extended conformation of the WW domains but also indicate the presence of transient contacts due to interdomain mobility (Fig. 1B and Supplementary Fig. 6). Finally, SAXS data show very good agreement

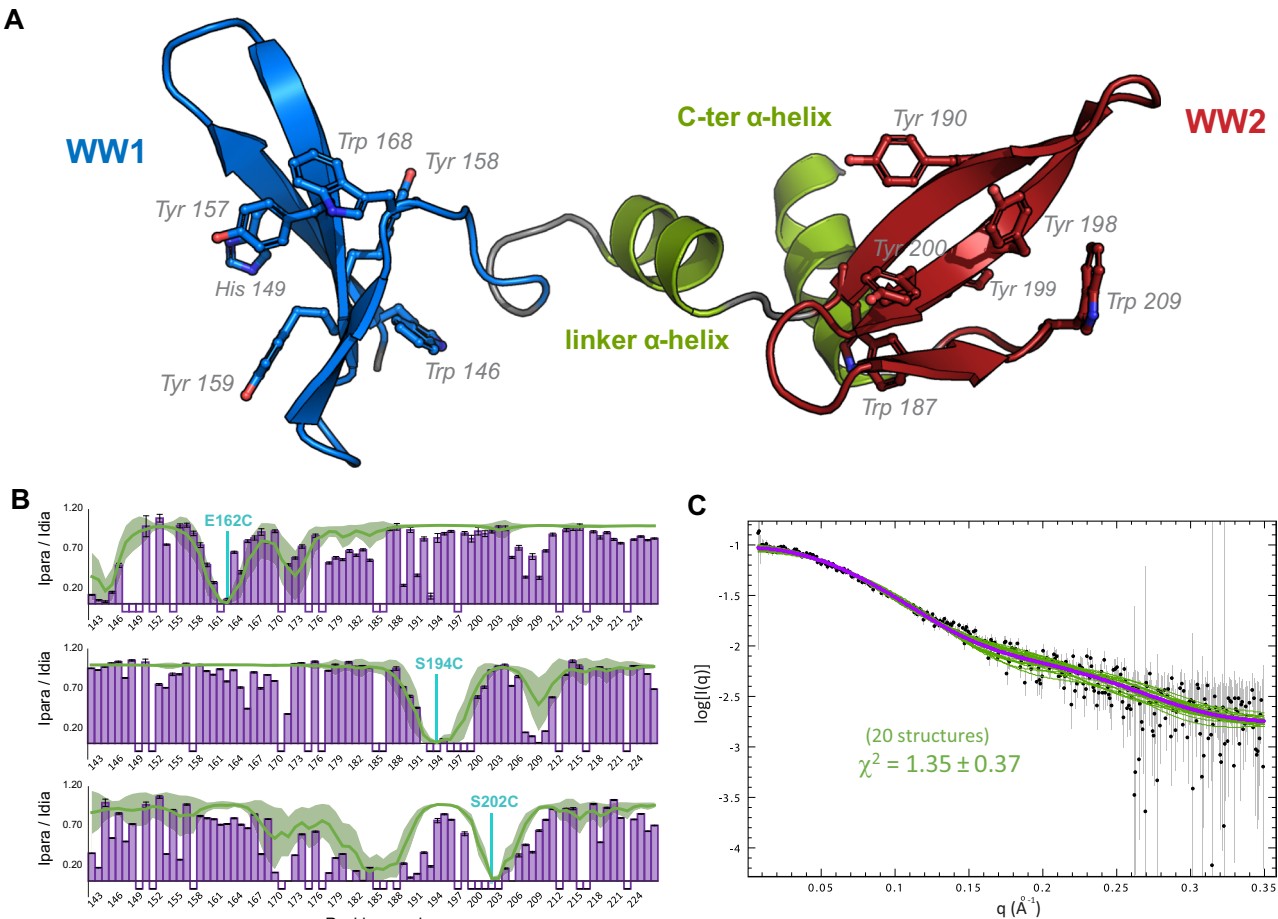

**Fig. 1 | Structure of PRPF40A tandem WW domains. A** Cartoon representation of a representative structure from the calculated ensemble for PRPF40A WW12; the aromatic residues, essential for the folding and ligand interaction, are shown in sticks. **B** Bar charts representing the intensity ratios for each amine signal in the $^1$H-$^{15}$N HSQCs between the paramagnetic and diamagnetic states of the IPSL-labeled proteins; experimental errors are shown as gray bars. Three different positions were selected for labeling: 162, 194, and 202. The green line and the shadow represent the calculated intensity ratios based on the structure of the PRPF40A WW12 and the deviation of the value in the 20-conformer ensemble. Negative values indicate no data for those residues. **C** Small Angle X-ray Scattering of PRPF40A WW12 protein (black dots indicate scattering intensity values and gray lines the measurement errors), the fitting of the calculated structural ensemble (20 green lines), and the averaged fitting of the ensemble (purple line). Source data are provided as a Source Data file.

($\chi^2 = 1.35$) with the ensemble of structures calculated by NMR (Fig. 1C and Supplementary Table 2). In contrast, the previously reported structure is inconsistent with these data (Supplementary Fig. 6). Therefore, our comprehensive analysis indicates that the PRPF40A WW tandem domains adopt an extended arrangement with a certain degree of interdomain mobility.

### PRPF40A recognizes a specific proline-rich motif in the SF1 C-terminal region

Few interaction studies have characterized the binding of PRPF40A WW domains to proline-rich sequences. Most of these studies only explored the binding of the first WW motif [56,60], while other studies, that used C-terminally truncated versions of the WW tandem [38,61], were unable to identify a specific high-affinity motif. In order to perform a systematic analysis of PRPF40A ligands, we first determined the minimal length of polyproline peptides (6–29) required for binding simultaneously to both WW domains by ITC. These data show that a peptide with 16 consecutive proline residues has the highest apparent binding affinity and equimolar stoichiometry for WW12 (Fig. 2A, Supplementary Fig. 7, and Supplementary Table 3).

We then selected different peptides of 16 residues in length in the C-terminal region of SF1 and performed ITC experiments to identify a potential high-affinity ligand in SF1 (Fig. 2B, Supplementary

Fig. 7 and Supplementary Table 3). The peptides containing spaced proline residues showed the lowest or no affinity; peptides with two stretches of 3–4 proline residues had a higher affinity ($K_d \approx 10\,\mu$M); but when any other residue breaks those polyproline stretches, the affinity drops ($K_d \approx 20$–$30\,\mu$M); except in the case of the highest affinity peptide (PPLPGAPPPPPPPPPP, residues 575–590; $K_d \approx 1\,\mu$M; hereafter named as SF1 PRPF40A WW binding site or SF1$_{WWbs}$). This high-affinity peptide harbors a leucine, breaking the polyproline tract. In fact, a PPLP motif was previously reported as a high-affinity peptide for the first WW domain of PRPF40A [61], suggesting that the polyproline region at the C-terminus of the peptide interacts with the second WW domain.

NMR titration of the $^{15}$N labeled WW12 tandem with the unlabeled SF1$_{WWbs}$ peptide shows binding in slow exchange regime, consistent with the low micromolar affinity (Fig. 2C and Supplementary Fig. 8). The most perturbed amino acid residues belong to the β-sheets of both WW domains, the canonical binding interface. In addition, the helical regions are also perturbed, suggesting either their involvement in the binding or a rearrangement of the tandem conformation upon binding of the ligand (Fig. 2D). When comparing the binding mode of SF1$_{WWbs}$ (Fig. 2D, bottom, green bar chart) with the polyproline 16-mer peptide (Fig. 2D, second row, blue bar chart), differences are observed between the spectra of the bound

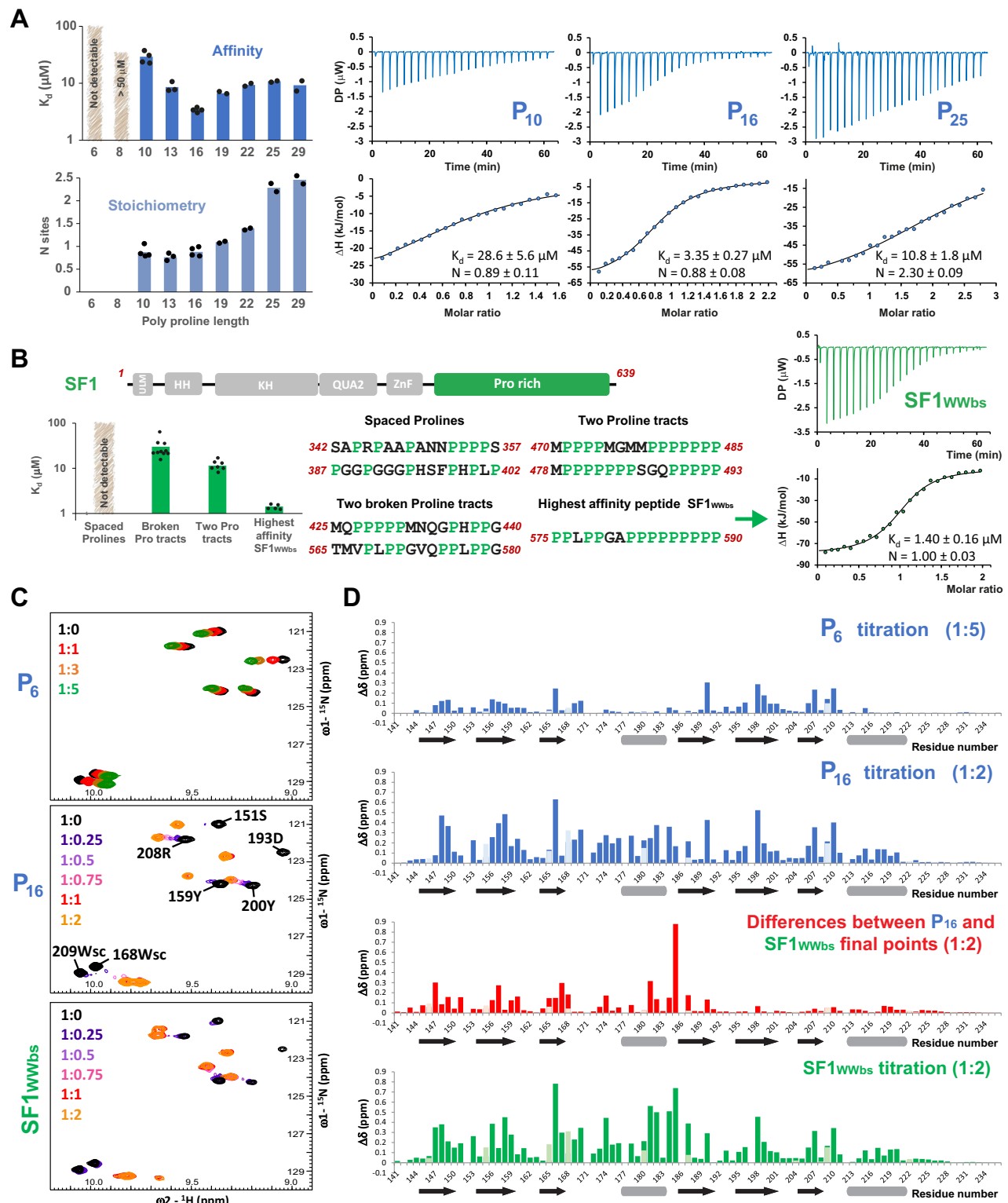

states (Fig. 2D, third row, red bar chart). The WW2 signals perfectly overlap, suggesting that the second domain interacts with the C-terminal region of both peptides, which share a polyproline sequence. The WW1 signals are slightly differently perturbed (meaning different endpoints between both titrations) due to the interaction with either the polyproline or the PPLP motifs at the N-terminal part of the peptides. In addition, there are some chemical shift differences of the residues in the helical linker, which could be explained by the different conformation that the peptides may have in the middle region, affecting the overall

arrangement of the tandem WW domains. This same region shows no perturbation in the titration with the shorter 6-mer peptide (Fig. 2D, first row, blue bar chart). Overall, these data suggest that the perturbation in the helical regions is not related to direct binding but to a change in the conformation that the WW domains adopt to accommodate the different long peptides (16-mer polyproline or SF1WWbs peptide) and when the peptide is short (6-mer polyproline), there is individual binding of a peptide molecule to each WW domain, without affection of the linker and therefore the tandem conformation.

**Fig. 2 | Analysis of the interaction between PRPF40A WW12 with different proline-rich peptides. A** Bar chart indicating the affinity and stoichiometry values between PRPF40A WW12 and GB1-polyproline peptides of different lengths measured by ITC; the individual data points are shown as black dots. Three selected titrations are shown on the right, displaying the fitting line; full ITC analyses in Supplementary Table 3 and Supplementary Fig. 7. Note that the peptides were in the cell and the WW tandem in the syringe, so N sites >1 represent more than one WW tandem binding to one peptide. **B** Bar chart (left) showing the average affinity and individual data points (black dots) of different types of 16-mer peptides contained in the C-terminal region of SF1 (up). The four types of peptides are described in the middle and in Supplementary Fig. 7. The ITC titration and fitting of the peptide showing the highest affinity (SF1$_{WWbs}$) is on the right. All ITC titrations in

(**A**, **B**) were repeated for each peptide at least twice. **C** Zoomed view of the $^1$H-$^{15}$N HSQC spectra showing the titration points of different peptides (P6—up, P16—middle, and SF1$_{WWbs}$—down) to PRPF40A WW12. In the middle spectrum, amide signals are labeled; sc refers to the Nε-Hε correlation of the tryptophan side chain. Full spectra in Supplementary Fig. 8. **D** Chemical shift perturbation charts for the same titrations as in **C**: with P6 peptide (up blue), with P16 peptide (down blue), with SF1$_{WWbs}$ (green), and the comparison between the bound states (1:2) with the P16 and with SF1$_{WWbs}$ peptides (red). Secondary structure elements are depicted in the x-axis using black arrows for β-strands and gray bars for α-helices. CSP data from backbone amide signals are shown in dark color bars and CSP data from the side chains in light colors (blue, red, and green) for the same amino acid position. Source data are provided as a Source Data file.

## Solution structure of the PRPF40A tandem WW domains in complex with SF1$_{WWbs}$

Chemical shifts of WW12 and the SF1$_{WWbs}$ peptide in the bound state were completely assigned, except for the middle region of the peptide (Pro 583–Pro 585), whose signals are broadened beyond detection. We found that the complex is more compact and presents fewer dynamics compared to the apo WW12. (i) RDC data fitted to the individual WW domain structures (apo) show a better agreement between both calculated alignment tensors for the WW domains, and (ii) $^{15}$N NMR relaxation data show less flexibility for the linker region in the complex (Supplementary Fig. 9).

Next, we determined the NMR structure of the complex (Fig. 3A and Supplementary Table 4) by the combination of the RDC data for PRPF40A and NOE-derived distance restraints, including 261 intermolecular NOEs (Supplementary Fig. 10), from complementary labeling schemes ($^{13}$C,$^{15}$N labeled PRPF40A and unlabeled SF1 and vice versa). In addition, we validated the structures of the complex using SAXS and NMR PRE data (Supplementary Table 2 and Supplementary Fig. 11). Both types of orthogonal data in solution agree with the calculated structural ensemble.

The calculated ensemble shows low coordinate r.m.s.d. (0.8 Å for the backbone and 1.2 Å for heavy atoms) for the folded region of the complex (PRPF40A residues 146-221 and SF1 residues 575-582, 586-590), indicating a compact arrangement of the WW domains in solution (Supplementary Fig. 12). The coordinate r.m.s.d. of the individual WW domains bound to the peptide is much lower (Supplementary Fig. 12), while the linker region of the peptide between the two binding sites is not well-defined due to the absence of experimental restraints. This may be due to the presence of an exchange process (probably proline cis-trans isomerization) in this part of the peptide, which is free and not interacting with PRPF40A. Note, that all visible prolines are in trans conformation according to the NOE pattern and Cβ and Cγ chemical shift values.

The structure shows that the first WW domain recognizes the PPLP motif at the N-terminus of the peptide while the second WW domain interacts with the C-terminal end of the proline-rich peptide in a similar arrangement as the WW1, as we previously suggested. There is a canonical recognition of the proline-rich peptide for both domains where the WW fold creates a hydrophobic patch with two defined grooves along the β-sheet that accommodates the peptide (Fig. 3B). The aromatic residues exposed from the WW domain surface of WW1 (His 149, Tyr 157, Tyr 159, Trp 168) and WW2 (Tyr 190, Tyr 198, Tyr 200, Trp 208) directly interact with the peptide positions 1 (Pro 575 and Pro 587), 3 (Leu 577 and Pro 589) and 4 (Pro 578 and Pro 590), respectively. The recognition involves extended hydrophobic interactions and some polar contacts (carboxyl groups of Pro 575/587 with hydroxyl of Tyr 157/Tyr 198 and carboxyl groups of Pro 756/Pro 588 with Hγ protons of Ser 166/Ser 207, Supplementary Fig. 13). WW1 displays unique recognition features. Most notably, Leu 577 of the peptide ligand is inserted in a large cavity created by the β1- β2 loop of WW1 (Supplementary Fig. 14), and the following proline, Pro 578, together with Pro 582 in the ligand sandwich the indole ring of the WW1 Trp 168. This

interaction seems supported by the extended conformation of the poly-proline peptide, which is a result of the recognition of the PPLG and PPPP motif in the peptide ligand by WW1 and WW2, respectively. To enable the tryptophan sandwiching by Pro 578 and Pro 582 the intervening Pro 579, Gly 580, Ala 581 residues adopt specific backbone conformations. This allows the peptide to wrap around the indole ring and further stabilizes the interaction by polar contacts of the backbone carboxyl groups of the peptide with the Trp Hε1 of the ring (Fig. 3C). For the WW2 domain, the β1– β2 loop has a different sequence and charge (instead of Pro 152 there is Asp 193), probably better accommodating the shorter hydrophobic side chain of Pro 589, which stacks against Tyr 200 (Supplementary Fig. 14). In addition, the proline sequence finishes with Pro 590, which interacts with the Trp 208 indole ring, and there is no available residue in the peptide to stack on the other side of the ring, although similar backbone salt bridges are found between carboxy groups of the peptide and the Hε1 proton of the indole ring.

In summary, the structure of the WW tandem of PRPF40A in complex with SF1 peptide shows the extensive interactions of the peptide with both WW domain surfaces. These interactions involve the canonical interface of the WW domains (WW1 binding to PPLP and WW2 to PPPP, respectively), but also reveal an unprecedented mode of recognition involving tryptophan sandwiching by two proline residues in the PPGAP motif.

## Binding of PRPF40A tandem WW domains to SF1 and SF3A1 proline-rich regions

To explore how the binding occurs with the close to full-length SF1 version, we performed ITC and NMR titrations using several longer constructs. First, we divided the proline-rich C-terminal region of SF1 in four parts (A-D, where C contains the high-affinity 16-mer WW$_{bs}$) (Fig. 4A) and tested their binding to WW tandem. All shorter SF1 constructs (ProA- ProD) showed binding by NMR but displayed different affinities as they presented various exchange regimens (Fig. 4B and Supplementary Fig. 15). ITC data also confirmed these different affinities (Fig. 4C, Supplementary Fig. 16 and Supplementary Table 3): the region containing the highest affinity motif SF1$_{WWbs}$ has a comparable apparent binding affinity to the isolated 16-mer, while the other regions show lower affinity, consistent with their different proline content. For example, the very C-terminal region contains two proline tracts separated by around 20 residues, which shows similar apparent affinity ($K_d$ around 10 μM) as the shorter 16-mers with also two complete proline tracts (Fig. 2B). When analyzing extended SF1 regions we observed that the apparent binding affinities remained similar to the SF1$_{WWbs}$ peptide but with higher stoichiometry (similar to the observation with the longer polyproline peptides) (Fig. 4C, Supplementary Fig. 16 and Supplementary Table 3). This indicates multiple binding events to the different proline tracts within SF1, the optimal SF1$_{WWbs}$ region, and suboptimal sites with an average stoichiometry of 3 PRPF40A tandems in the complete C-terminal region of SF1.

PRPF40A has also been linked to complex A, where SF1 is no longer present[39]: SF1 is not considered a component of the U2 snRNP in

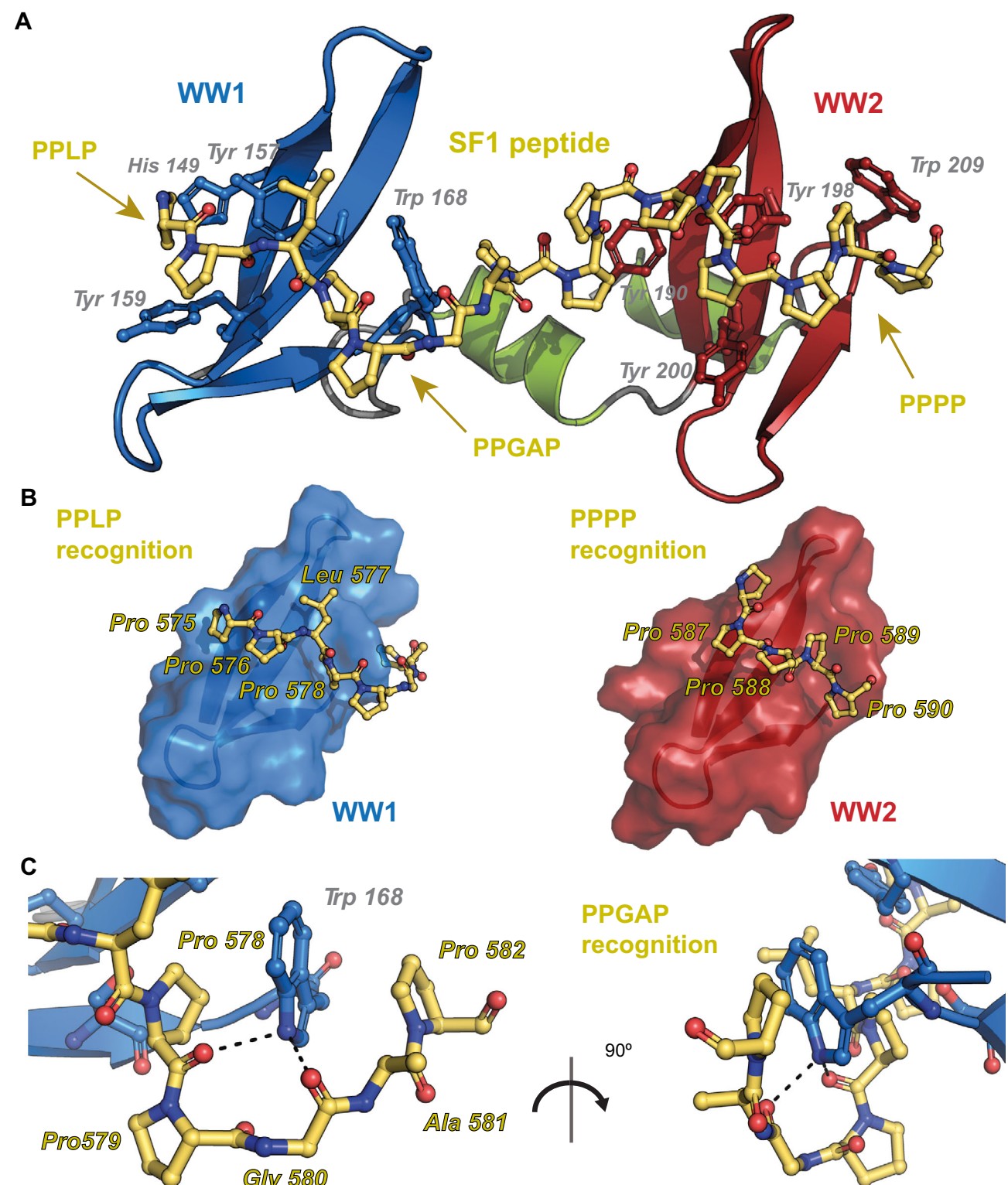

**Fig. 3 | Structure of PRPF40A tandem WW domains in complex with SF1$_{WWbs}$ peptide. A** Cartoon representation of WW12 tandem of PRPF40A bound to the SF1$_{WWbs}$ peptide (PPLPPGAPPPPPPPP, shown in sticks), indicating the recognition of the three segments in the peptide and the aromatic residues involved in the interaction. **B** Cartoon and surface view of the WW domains of PRPF40A, in the same orientation (WW1 left, WW2 right), with the interacting SF1 peptide depicted in sticks. The amino acids recognized by the canonical interface are labeled. **C** Zoom view of the stick representation of Trp 168 (WW1) sandwiched by the PPGAP motif of SF1: this Pro-Trp-Pro stacking interaction has not been observed in other WW domain interactions with proline-rich peptides.

complex A, but it has been shown to promote the recruitment of U2 particle to the 3′SS[15]. The U2 snRNP component, SF3A1, has been shown to interact with PRPF40A[62]. SF3A1 has mainly two proline-rich regions, one of them containing the high-affinity motif of WW1 (PPLP)

(Fig. 4D). We performed ITC titrations of this SF3A1 region (364–413) with the WW tandem of PRPF40A, obtaining comparable binding in the low micromolar range, although with lower apparent affinity than SF1$_{WWbs}$ ($K_d \approx 10\ \mu M$). This peptide contains three main proline-rich

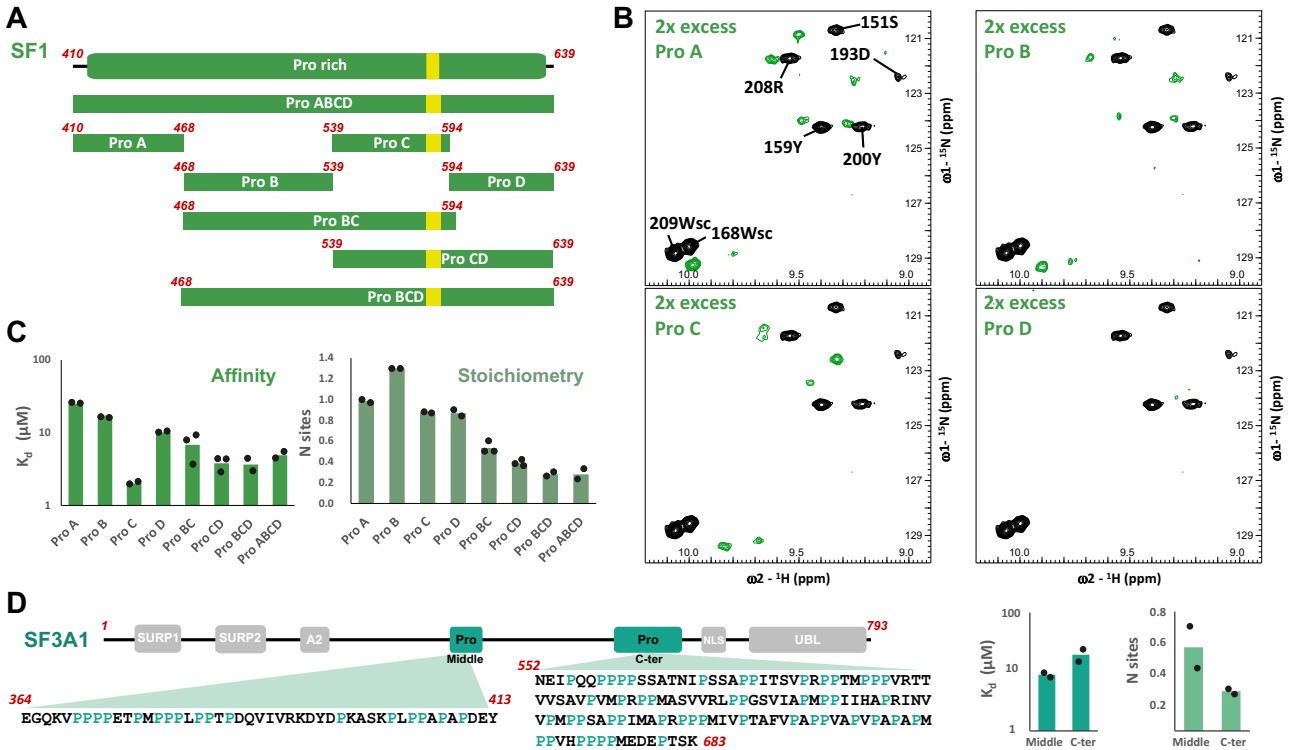

**Fig. 4 | Binding of PRPF40A WW domains to splicing factors SF1 and SF3A1.**
**A** Scheme representation of the different SF1 constructs utilized for the characterization of the binding of PRPF40A tandem to the C-terminal region of SF1. The high-affinity motif (WW$_{bs}$) for the tandem is shown in yellow. **B** Zoomed views of the $^1$H-$^{15}$N HSQC spectra overlay of free PRPF40A WW tandem (black) and titrated with two-fold molar excess of the shorter SF1 constructs Pro A to Pro D (green). In the first spectrum, amide peaks are labeled; sc refers to the Nε-Hε correlation of the tryptophan side chain. Full spectra in Supplementary Fig. 15. **C** Summary graph of the isothermal titration calorimetry results for the dissociation constant ($K_d$) and stoichiometry calculation (N site) for the titrations of PRPF40A with the SF1 constructs shown in **A**. Averaged values are represented by the bars and individual data are shown as black dots. **D** ITC results (right) for the titration of the two SF3A1 constructs (shown in left) with the PRPF40A WW tandem; averaged values (bars) and individual data points (black dots) are shown in the graphs. All ITC titrations in (**C**, **D**) are done at least in duplicates. Note that the SF1/SF3A1 constructs were in the syringe and the WW tandem in the cell so N sites <1 represent more than one WW tandem binding to one peptide. Source data are provided as a Source Data file.

regions, the PPLP motif and the middle one; complicating the binding analysis and interpretation as multiple binding events may occur. In any case, the absence of the PPLPPGAP sequence may be an explanation of the lower affinity this SF3A1 region shows compared to the SF1 constructs containing the SF1$_{WWbs}$ sequence. A second, longer proline-rich region in SF3A1 (552–683) displays lower apparent affinity and higher stoichiometry, probably indicating multiple low-affinity binding events (Fig. 4D, Supplementary Fig. 16 and Supplementary Table 3).

In summary, the WW tandem of PRPF40A recognizes a high-affinity motif in SF1 or SF3A1 but also shows some promiscuous binding to suboptimal proline-rich sequences. We note that the two WW domains are able to simultaneously interact with separated proline-rich sequences, but that the affinity is still directed by the different motifs they contain (PPLPPGAP, PPLP, PPPP). The presence of other lower affinity sequences within the same protein target could boost the interaction of both proteins through avidity effects and could preliminary load PRPF40A on SF1 until the optimal sequence is reached.

**Intramolecular binding of an N-terminal proline-rich region to the PRPF40A WW domains**
We noticed the presence of regions enriched in prolines in the N-terminal region of PRPF40A (59–127). To analyze their possible roles, we studied a PRPF40A construct containing the complete N-terminal region (residues 1–310), which comprises these proline-rich motifs and the WW tandem (Fig. 5A). The comparison of $^1$H-$^{15}$N HSQC

spectra of the N-terminal region with the WW12 construct alone showed the typical $^1$H chemical shift dispersion of secondary structured elements only for the WW domain tandem signals (Supplementary Fig. 17). Surprisingly, many residues in the WW domains show chemical shift changes in a similar way as observed during titrations with proline-rich peptides. This suggests that the N-terminal proline-rich region may establish intramolecular interactions with the WW domain tandem (Supplementary Fig. 1). In fact, a shorter version of the construct including only the proline-rich region and the WW domains (N-ext-WW12, residues 56–236) has comparable NMR spectra (Supplementary Fig. 17), demonstrating that the cause of the perturbation of the WW domains is located indeed N-terminal of the tandem.

While only partial backbone chemical shift assignments of this construct (69% completeness), N-ext-WW12, could be possible due to a highly repetitive sequence, a complete assignment of the methyl groups was achieved. We then performed a titration with the high-affinity peptide, SF1$_{WWbs}$, by NMR. A comparison of $^1$H-$^{15}$N and $^1$H-$^{13}$C correlation spectra of the N-ext-WW12 construct free and in complex with SF1$_{WWbs}$ (at 1:2 molar ratio) (Supplementary Fig. 18) shows the typical shifts on the WW tandem region due to SF1$_{WWbs}$ binding, but also revealed high perturbation on two regions at the N-terminal part, narrowed in two motifs (PPVP, residues 78-81 and PALPP, residues 123-127) (Fig. 5B). This suggests that these two motifs are involved in the intramolecular interaction with the WW domains in the free PRPF40A, and are released upon binding with the SF1 peptide. To further support this conclusion, we prepared a mutant version of the N-ext-WW12 construct, changing these two motifs to alanine residues, and

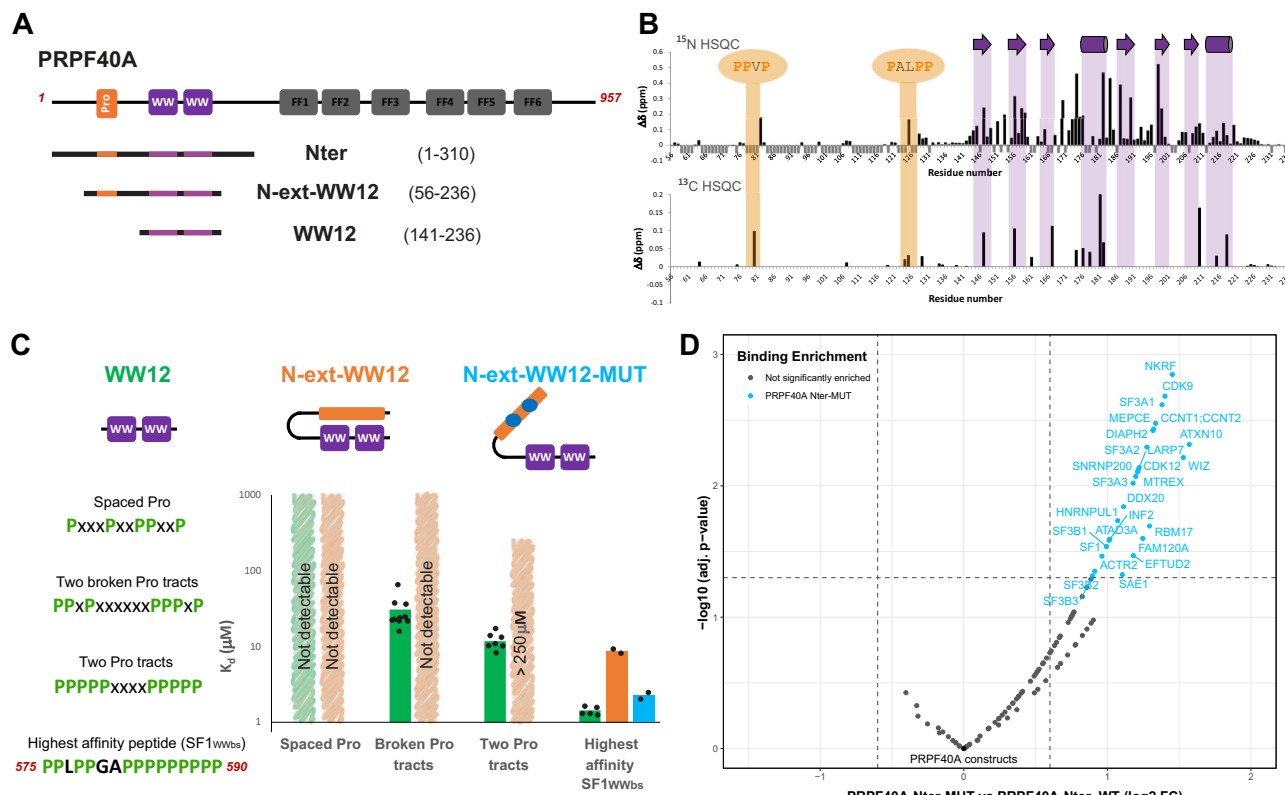

**Fig. 5 | The N-terminal region of PRPF40A improves the selectivity of the WW domains. A** Different extended constructs of the N-terminal region of PRPF40A. **B** NMR titration analysis of the N-terminal extended WW12 construct with the high-affinity peptide from SF1 (SF1$_{WWbs}$): two chemical shift perturbation plots are shown for the comparison of $^1$H-$^{15}$N (top) and $^1$H-$^{13}$C HSQC spectra between the free and bound (1:2) states (spectra shown in Supplementary Fig. 18). The most perturbed region corresponds to the WW12 binding interface as in the WW12 titrations; but in the N-terminal extension, two regions appear also affected (due to release from WW12 intramolecular binding). Negative gray bars indicate no data. **C** Bar chart showing the affinities calculated by ITC for the different types of SF1 peptides with the three WW12 versions (isolated WW domains—green, N-terminal extended

WW12 construct—orange, mutated version of the N-extWW12—blue). Bars indicate averaged values while individual data points are shown as black dots; all ITC titrations were done at least in duplicates for each peptide. **D** Plot comparing interactors of GFP-tagged PRPF40A-Nter-WT vs PRPF40A-Nter-MUT construct. In the PRPF40A-Nter-MUT construct, the proline-rich sequence was substituted by a polyalanine stretch. To remove background, only proteins that are significantly enriched for the GFP-tagged WT- or Mut-PRPF40A constructs compared to GFP alone are considered. Significantly altered protein interactions are shown in blue (log2 FC > log2(1.5), adjusted p-value < 0.05), non-significant binding differences are shown in gray. Adjusted *p* values (two-sided) were determined by linear modeling using the limma package. Source data are provided as a Source Data file.

observed clear shifting of NMR signals towards the free WW12 spectrum in the short version of the tandem. This corroborates that mainly these two proline-rich motifs govern the intramolecular interacting region (Supplementary Fig. 19).

### Intramolecular interactions provide specificity for WW domain peptide recognition

The intramolecular interaction seems to mimic the binding of proline-rich peptides to the WW tandem, displaying a possible autoinhibitory role. We therefore analyzed the effect of the N-terminal autoinhibition on the binding to different intermolecular polyproline ligands. We performed NMR and ITC titrations with several peptides. For the high-affinity peptide (SF1$_{WWbs}$, $K_d \approx 1\,\mu$M) the methyl signals of the extended region (Val 80 and Val 129), which in the apo version interact intramolecularly with the WW12, are released from the tandem (their chemical shifts are shifted towards random coil values) upon binding already at the 1:1 ratio. This means that the SF1 peptide is able to displace the intramolecular interaction with the N-terminal extension and bind to the WW domains (Supplementary Fig. 20). On the other hand, a similar titration experiment with a moderate affinity ligand (SF1$_{470-485}$, MPPPPMGMMPPPPPPPP, $K_d \approx 10\,\mu$M) shows that the methyl signals of the extension remain bound at a certain percentage even when saturation up to 1:4 ratio This indicates that this peptide cannot fully displace the intramolecular motifs to bind the WW domains and

thus, the intramolecular autoinhibition is able to compete with the moderate affinity peptide for the WW12 (Supplementary Fig. 20).

The previous results suggest that the presence of the N-terminal extension can modulate the affinity for different proline-rich peptides, and compete with those that are not optimal. To prove this further, we compared the ITC titrations of three different peptides for WW12 or N-ext-WW12 constructs. We found that the presence of the extension reduces the affinity for proline-rich peptides in all cases, due to the competitive effect of its presence. In the case of the high-affinity peptide, SF1$_{WWbs}$, the affinity drops from a $K_d$ of 1.4–8.8 $\mu$M while for the other peptides, which showed $K_d$ values beyond 10 $\mu$M, now the affinity is too low to be quantified by ITC and is estimated to be in the order of hundreds of $\mu$M (Fig. 5C, Supplementary Table 3 and Supplementary Fig. 21). To further confirm this analysis, we titrated SF1$_{WWbs}$ and the mutant version of the N-ext-WW12 construct (changing the PPVP/PALPP motifs for alanine residues). This restored the binding affinity to be comparable to the isolated WW tandem, demonstrating that the two proline-rich regions are responsible for mediating the affinity of the WW domains (Fig. 5C, Supplementary Table 3, and Supplementary Fig. 21).

### Intramolecular autoinhibition mediates binding specificity in cells

Next, we investigated to what extent the intramolecular interaction between the WW domains and the proline-rich sequences modulates

PRPF40A activity in cells. We compared the interactome of the WW domains with and without the regulatory N-terminal sequences using immunoprecipitation coupled to quantitative mass spectrometry (SILAC IP-MS). To this end, we created vectors that express GFP-tagged PRPF40A N terminal region (PRPF40A-Nter-WT), a mutated version in which the PPVP/PALPP motifs are changed to alanine residues (PRPF40A-Nter-MUT) or GFP alone as a control. Upon expression of the constructs in Hela cells and GFP-immunoprecipitation, co-purified interactors were analyzed by quantitative mass spectrometry (Supplementary Fig. 22A). In comparison to GFP alone, both PRPF40A-Nter constructs presented interactions with proteins involved in cytoskeletal organization, consistent with previous reports[36] (Supplementary Table 5 and Supplementary Data). Importantly, we identified several U2 snRNP-related proteins, namely SF1 and the SF3A and SF3B complexes, which further proves that PRPF40A is able to interact with U2 snRNP through its WW domains (Supplementary Fig. 22B, C, Supplementary Table 5 and Supplementary Data). PRPF40A is proposed to interact with the U1 snRNP with its FF domains, which are not included in the PRPF40A-Nter constructs, and consequently, no U1 snRNP-associated protein was found. Interestingly, we find interactions with several RNA polymerase II-associated proteins like CDK9, CCNT1/2, LARP7, and MEPCE.

When comparing the PRPF40A-Nter-WT and -MUT constructs, the mutated version displays generally more promiscuous binding (Fig. 5D, Supplementary Fig. 22A). PRPF40A-Nter-MUT specific binding is most enriched for several proteins with Pro-rich regions, also including those harboring sub-optimal PPLP or PPPP motifs (Supplementary Table 5), suggesting that PRPF40A autoinhibition prevents unspecific binding to these moderate affinity motifs. On the other hand, SF1 binding enrichment is rather moderate, consistent with the presence of the high-affinity proline-rich motif (PPLPPGAP/PPPP), which is not present in any of the other proteins (the closes sequence is in DIAP2: PPLPGGAPLPPPPP).

In summary, these results support our biophysical observations that the intramolecular interactions in PRPF40A generally reduce the binding affinity to proline-rich ligands, but have an important role in enhancing binding selectivity by discriminating high and low-affinity ligands.

## Discussion

### Structure of PRPF40A tandem WW domains and proline-rich peptide recognition

Our structure of the PRPF40A WW tandem and experimental validation using various techniques show two domains acting as independent modules separated by a hinge and a short helix; in addition, we have shown the presence of an extra C-terminal α-helix that runs parallel to the linker helix. This structure differs from a previously reported structure[38], in which a truncation of the C-terminal helix or erroneous automated NOE assignments may have impaired the structure determination. Using two orthogonal solution techniques (NMR and SAXS), we have validated our ensemble in contrast to the truncated published model.

Several examples of WW tandem structures have been reported with various degrees of linker flexibility and therefore, inter-domain interactions. Highly flexible linkers are found, for example, in the human FBP21[63] and YAP proteins[64], or the fly Su(dx) WW3-4 tandem from the NEDD4 family[65]; while rigid tandems are present in the yeast homolog of PRPF40A, Prp40, which shows a fix tandem structure with a continuous linker helix from WW1 to WW2 that rigidifies the system[66]. The surprising difference between the two PRPF40 homologs can be explained when comparing their linker sequences: the presence of a proline residue in the linker of the human version breaks the continuity of the helix (Supplementary Fig. 23). This disruption of the helix enhances the conformational dynamics of the linker and results in the differential domain arrangement. A flexible tandem

would easily accommodate peptides of different compositions, like in the case of PRPF40A binding to the polyproline 16-mer vs the SF1$_{WWbs}$ peptide, with a local rearrangement of the tandem orientation.

In certain cases, the binding to the peptide promotes interaction between the two WW domains in the tandem as it has been observed for the WW domains of PLEAKHA7 upon binding to PDZD11[67]; or additionally, the presence of one WW domain may have a chaperone activity for the other, promoting its folding, as in the case of WW domains of WWOX[68]. This exact case was proposed for PRPF40A WW domains, where WW1 would promote the folding of WW2[38]. However, our data demonstrate that WW2 can be folded independently of WW1 (Supplementary Fig. 3).

Our structure of the WW tandem of PRPF40A in complex with SF1 peptide reveals extensive interactions of the peptide with both WW domains, which involve the canonical interface of the WW domains (WW1 binding to PPLP and WW2 to PPPP, respectively). Notably, the structure also reveals an unprecedented mode of recognition involving tryptophan sandwiching by two proline residues in the PPGAP motif of the peptide, which connects the WW1 and WW2 binding motifs and greatly enhances the specific recognition and binding affinity.

### Intramolecular autoinhibition proofreads the binding selectivity of WW domains

WW domains normally exhibit some preference for certain types of proline-rich motifs. In fact, an initial classification of WW domains was proposed based on this[53]. But at the same time, WW domains are rather promiscuous as individual domains, and only when they are acting in combination with other WW domains, the selectivity for certain proteins increases[54,55,67]. The affinity of individual WW domains for proline-rich peptides is normally modest with a dissociation constant typically in the range of hundreds of micromolar[48] (Fig. 6A). This is also the case of PRPF40A individual WW domain studies, mainly WW1, whose affinities for proline-rich peptides range from 150 to 600 μM[56,60,69]. Our NMR titration data for individual domains (Supplementary Fig. 24) also corroborate this moderate affinity for both PRPF40A WW domains.

When WW domains are combined in tandem arrangements, they cooperate to bind longer peptides containing binding motifs for both WW domains, yielding higher affinity (Fig. 6A). The extent of this increase depends on mainly two factors, the distance between the two binding sites in the peptide and the coupling between the two domains[70]. In the case of the coupled WW domains that exhibit inter-domain interactions upon binding, the increase of affinity is much higher, reaching dissociation constants in the nanomolar range, for example in the case of the PY binding WW domains of KIBRA and MAGI proteins[54]. For PRPF40A, no interdomain interaction or strong domain coupling is observed and therefore, the affinity is lower. For the interaction with SF1 WW$_{bs}$ the dissociation constant is at low micromolar range. This implies that the promiscuity for other similar sequences is higher, in fact, our data show that the affinity for other sequences within SF1 is only 5–10 times lower.

A surprising finding of our study is that an N-terminal region of PRPF40A, which is conserved among metazoan but not in yeast (Supplementary Fig. 1) contains proline-rich regions and can mediate an intramolecular interaction with the WW domains mimicking the target peptide. This intramolecular interaction competes with the different intermolecular ligands for binding the WW domains and reduces the affinity for all targets. However, we have found that in those cases with moderate affinity, the presence of the N-terminal extension heavily reduces the binding, estimated in the higher micromolar range. On the contrary, the effect of the extension on the binding to the high-affinity peptide SF1$_{WWbs}$ is less severe and the dissociation constant is still in the low micromolar range (less than 10 μM). Indeed, our immunoprecipitation data clearly shows that the presence of the N-terminal region reduces the binding of all interacting

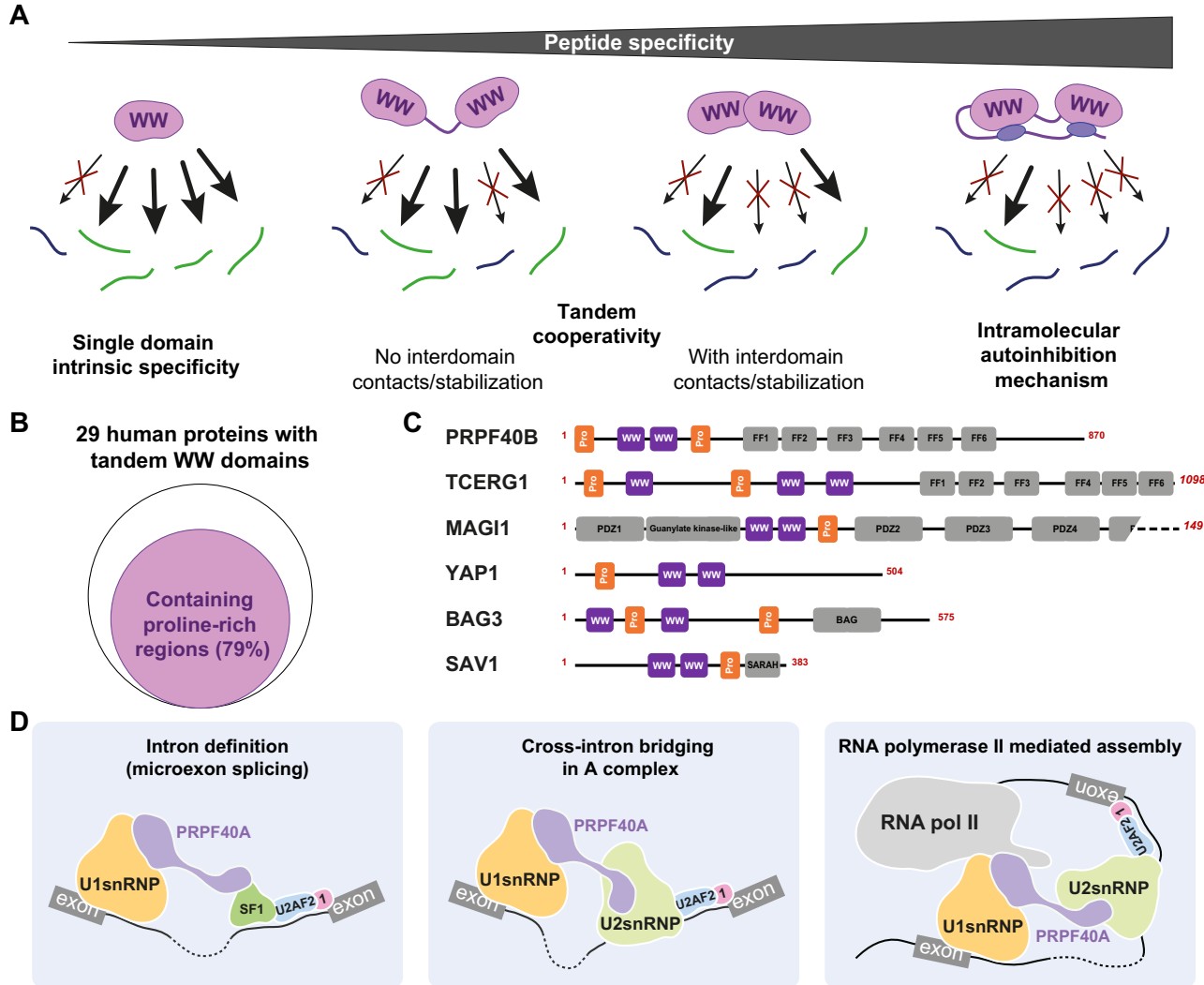

**Fig. 6 | Binding selectivity of WW-containing proteins and potential roles of PRPF40A during splicing. A** Selectivity mechanisms of WW-containing proteins. The different layers of WW domain specificity towards the recognition of Pro-rich peptides start with the intrinsic preference for certain motifs. The selectivity gets increased with these domains combined in tandems and it is even higher if the domains show interdomain contacts upon binding to the right sequences. Finally, the additional layer of selectivity comes from the role of the regulatory regions nearby the WW domains that are able to compete out those suboptimal binding events. **B** Diagram showing the percentage of all human proteins containing a tandem of WW domains that also have proline-rich sequences and examples of some of those proteins as schematic domain representations (**C**). **D** Potential roles of PRPF40A in early spliceosome assemblies. PRPF40A role in bridging the branch point region of the intron and the 5′splicing site could be important for (1) the intron definition events, when the exon definition cannot happen, like in the microexon regulation (left), (2) reinforcement of the A complex bridging (middle) and/or (3) in those events that happen co-transcriptionally due to its shown interaction with the C-terminal domain of the RNA polymerase II (right).

proteins, but those that show higher affinity (SF1) display more moderate enrichment when the regulatory region is mutated.

We propose that this autoinhibitory regulation is important for scanning different proline-rich sequences and promotes the binding to the correct target, avoiding unspecific interactions. In fact, analyzing the sequences of all WW domain tandem-containing proteins in humans, we have found that more than half of them also possess proline-rich sequences N or C-terminally placed to the WW domains (Fig. 6B). This is the case of the homolog PRPF40B, and closed related protein TCERG1, the three MAGI proteins, HECW1/2, YAP1, BAG3, SAV1, FNBP4, WWP2, ITCH, NEDD4, ARHGAP27 and ARHGAP39 (Fig. 6C). In fact, our own data on PRPF40B shows that the presence of its N-terminal extension produces chemical shift differences in the NMR spectra (Supplementary Fig. 25). Whether the internal proline-rich sequences that these proteins have, act in a similar way as in PRPF40A still needs further analysis, but it is reasonable to

hypothesize that they may play a competing role to rule out unspecific interactions (Fig. 6A).

## The role of PRPF40A in early spliceosome assembly

Recent cryo-EM structures of the early stages of the spliceosome assembly enlighten our knowledge of the molecular details that drive the recognition of the splicing sites that are meant to undergo splicing and the early contacts that the two splicing sites make in these early steps. In yeast, structures of the U1 snRNP[71], E complex[32], pre-A complex[19], and A complex[31] incorporate Prp40 (yeast PRPF40A) as a constitutive component of the U1snRNP. However, only the C-terminal region of the protein (the FF domains) is visible, interacting with different regions of the U1snRNP. The WW domains are always missing in these cryo-EM structures, probably due to the intrinsic flexibility of the systems. Nevertheless, an interaction with the SF1 homolog, Msl5, is proposed or supported by mass spectrometry. In fact, the role of

**Table 1 | Sequences of the SF1 derived 16-mer constructs used in this work**

| | |
|---|---|
| SF1$_{342-357}$ | SAPRPAAPANNPPPPS |
| SF1$_{387-402}$ | PGGPGGGPHSFPHPLP |
| SF1$_{425-440}$ | MQPPPPPMNQGPHPPG |
| SF1$_{470-485}$ | MPPPPMGMMPPPPPPP |
| SF1$_{478-493}$ | MPPPPPPPSGQPPPPP |
| SF1$_{565-580}$ | TMVPLPPGVQPPLPPG |
| SF1$_{575-590}$ = SF1$_{WWbs}$ | PPLPPGAPPPPPPPPP |
| SF1$_{596-611}$ | MYAPPPPPPPPMDPSN |
| SF1$_{623-638}$ | MPPFGMPPAPPPPPPQ |

Prp40 was already described as a cross-intron bridging factor between 5′ and 3′splicing sites[29].

In metazoans, the exact role(s) of PRPF40A/B is unclear. Splicing is a general mechanism in higher eukaryotes, while otherwise residual in yeast (< 1% of the genes contain introns). Therefore, it is not surprising that metazoan splicing shows more complex regulation involving numerous splicing factors and degenerate *cis* regulatory motifs. One consequence of this is the significant presence of alternative splicing. The regulation of alternative splicing involves the interplay of common core components like the snRNPs and hundreds of splicing factors that affect different pre-mRNAs in various situations. PRPF40A/B may act more like alternative splicing factors rather than general core factors[40–42]. In fact, assuming that, as the yeast Prp40, human homologs are involved in cross-intron bridging through intron definition mechanism, their function would be important for those special cases in which the intron definition drives the recognition of the splicing sites, like in the microexon splicing[42] (Fig. 6D); while in higher eukaryotes the exon definition (cross-exon interaction of U1 and U2 snRNPs) is the most common recognition mechanism[72]. Alternatively, PRPF40A/B would be also important for establishing cross-intron connections once the splicing sites are recognized: PRPF40A/B are sometimes classified as U2 snRNP-associated components[35], and as we have shown, SF3A1, a core component of the U2 particle, contains a proline-rich sequence with a PPLP motif able to interact with the WW tandem, which could serve of anchor point for PRPF40A/B once SF1 has been released (Fig. 6D). Finally, PRPF40A has also been shown to interact with the CTD of RNA polymerase II[34], consistent with the detection of RNA polymerase II-associated proteins in our pull-down experiments. Moreover, the close homolog TERCG1 is considered a transcription elongation factor that copurifies with early spliceosomal complexes[73–75]. This suggests the possibility that PRPF40 homologs may participate in the 3′–5′ bridge connection in the context of co-transcriptional splicing (Fig. 6D).

In summary, our study established the molecular bases for the recognition of proline-rich sequences by PRPF40A and identified relevant ligand motifs with roles in early spliceosome assembly. The presence of intramolecular autoinhibitory interactions in more than half of all tandem WW domain containing proteins, suggests a general mechanism of how the promiscuous interactions are proofread to identify bona fide ligands to regulate WW mediated protein-protein interactions.

## Methods

### Protein expression and purification

PRPF40A isoform 1 sequences (N-ter 1–310, WW12short 141–223, WW12 141–236, WW12long 141–248, and N-ext-WW12 56–236) were amplified from a purchased DNA fragment (IDT) and cloned using NcoI and KpnI restriction sites into a pETM11 vector encoding 6xHis and TEV cleavage site as a N-terminal tag. SF1 (Pro A 410–468, Pro B 468–539, Pro C 539–594, Pro D 594–639, Pro BC 468–594, Pro CD 539–639, Pro BCD 468–639 and Pro ABCD 410–639) and SF3A1 (Middle Pro 364–413 and C-ter Pro 552–683) constructs were prepared in a similar manner using a pET24 vector with a 6xHis, GB1 and TEV cleavage site as N-terminal fusion cassette. 16-mer SF1 constructs (Table 1) and polyproline peptides (P$_6$, P$_8$, P$_{10}$, P$_{13}$, P$_{16}$, P$_{19}$, P$_{22}$, P$_{25}$, and P$_{29}$) were cloned in the same pET24 vector with GB1 tag by direct ligation of purchased oligonucleotides (Eurofins). All peptide sequences contained N and C-terminal common residues to avoid boundary effects on binding: TEVsite-GAMSGS-peptide-SGSN. Mutants of PRPF40A WW12 and N-ext-WW12 constructs were prepared by site-directed mutagenesis. The truncated version of PRPF40A WW12 (141-217) was a gift from Dr. María Macías (IRB, Spain). All used oligonucleotides were purchased from Eurofins Genomics and are shown in Supplementary Table 6.

Plasmids were transformed into *Escherichia coli* BL21(DE3) cells and proteins were overexpressed in Luria-Bertani (LB) broth for natural abundance samples or M9 minimal media supplemented with 2 g/l of $^{13}$C glucose and/or 1 g/l $^{15}$NH$_4$Cl for labeled samples. Typically, expression was induced in cultures at OD$_{600}$ = 0.6–0.8 with 0.5 mM IPTG at 22 °C for about 16 h and then cells were harvested and frozen at −20 °C or directly processed.

Resuspended cell pellets in 20 mM Tris pH 8.0, 500 mM NaCl, 10 mM imidazole, 0.5 mM TCEP, and protease inhibitors mix (SERVA) were lysed by cell disruption in a French press. Clear lysates, after centrifugation (at 18,000 g) and filtering, were loaded into Ni-NTA resin. The resin was then washed with 20 mM imidazole containing buffer, and the desired protein was eluted with 500 mM imidazole. PRPF40A constructs were then subjected to TEV protease treatment (homemade at a final concentration of ≈50 μg/ml), dialyzed overnight at 4 °C against an imidazole-free buffer, and then loaded again to Ni-NTA resin; desired digested proteins were obtained in the flow-through and wash fractions while uncleaved protein, TEV protease, and impurities remained bound to the resin. Cleaved PRPF40A proteins and GB1-tagged SF1, SF3A1, and polyproline constructs were further purified by size-exclusion chromatography on a HiLoad 16/60 Superdex 75 column (GE Healthcare) equilibrated with 20 mM sodium phosphate, pH 6.5, 100 mM NaCl and 1 mM DTT buffer. For the purification of PRPF40A WW12 in complex with SF1 peptide, cell pellets of recombinant expression for both proteins were mixed and processed together following the Ni-NTA−TEV digestion−reverse Ni-NTA−SEC-75 protocol; where the complex remained bound in a 1:1 stoichiometry throughout the whole purification (corroborated by NMR). Molecular weight of each sample was confirmed by mass spectrometry, purity was checked by PAGE-SDS, and protein concentrations were determined from the aromatic contribution to the UV absorbance spectra at 280 nm.

### NMR spectroscopy

NMR samples were prepared at concentrations ranging 100–1000 μM in buffer containing 20 mM sodium phosphate pH 6.5, 100 mM NaCl, 1 mM DTT, and 10% or 100% D$_2$O, and measured at 25 °C (except stated otherwise) in 900-, 800-, 600- and 500-MHz Bruker Avance III NMR spectrometers equipped with cryogenic triple resonance gradient probes. Topspin 4.1.0 (Bruker) software was used for NMR data acquisition. Spectra were processed with NMRpipe 8.9[76] and analyzed using ccpnNMR Analysis 2.5 software[77].

**Protein assignment and structure calculations.** NMR assignments of WW12 different constructs (WW12 free and bound to SF1$_{WWbs}$ peptide, WW12long, and N-ext-WW12) were obtained from triple resonance backbone experiments 3D HN(CA)CO, HNCO, CBCA(CO)NH, HNCACB, H(CC)(CO)NH-TOCSY and (H)CC(CO)NH-TOCSY[78]. Assigned $^1$H-$^{15}$N HSQC spectra for the three proteins are included in Supplementary Figs. 26, 27. Side chain assignments of the WW12 free and bound states

were completed using 2D TOCSY, 3D H(C)CH-TOCSY, and 2D (HB) CB(CGCD)HD and (HB)CB(CGCDCE)HE experiments[79] in 100% $D_2O$ buffer. For the structure calculation of the free WW12, a 3D $^1$H-$^{13}$C edited HMQC-NOESY implemented with ultrashort broadband cooperative pulses[80] (in 100% $D_2O$) and a 3D $^1$H-$^{15}$N edited HSQC-NOESY[81] (in 10% $D_2O$) were acquired using 120 ms of mixing time. Assignment of SF1$_{WWbs}$ peptide bound to WW12 was done after acquisition of 3D HNCACB and H(CC)(CO)NH backbone experiments (in 10% D2O), 2D TOCSY and 3D H(C)CH and (H)CCH TOCSYs (100% D2O) and 3D $^1$H-$^{13}$C edited HMQC-NOESY in samples containing double labeled SF1$_{WWbs}$ peptide and unlabeled WW12 protein at 35 °C. For the structure calculation of the complex, samples with one component (PRPF40A WW12 or SF1 peptide) double-labeled and the other in natural abundance were used for acquisition of various 3D NOESYs at 35 °C and with 150 ms of mixing time: $^1$H-$^{15}$N edited HSQC-NOESY (in 10% $D_2O$), $^1$H-$^{13}$C edited HMQC-NOESY (in 100% $D_2O$) and $^1$H-$^{13}$C edited $\omega_1$ filtered HMQC-NOESY (in 100% $D_2O$), also implemented with ultrashort broadband cooperative pulses, for unambiguous intermolecular NOE measurement.

NOESY spectra assignment was done first by automated assignment using ARIA 2.3[82] and then by manual curation. Dihedral constraints were derived from the chemical shift data using TALOSn software[83]. NOE-derived distance restraints together with dihedral restraints were implemented in ARIA2.3 software for calculation of 200 structures with torsion angle dynamics and using for the second cooling step the Log-Harmonic potential with 25 as weight values for both ambiguous and unambiguous restraints[84]. An ensemble of the 20 best structures was selected based on minimal energy criteria and subjected to further explicit water refinement using ARIA2.3. Final ensembles of free PRPF40A WW12 and in complex with SF1$_{WWbs}$ were analyzed with NMR-PROCHECK[85] and PSVS suite[86]. R.m.s. values were obtained from Molmol[87], and figures were drawn using Pymol (DeLano Scientific LLC, Palo Alto, USA).

**Relaxation experiments.** {$^1$H}-$^{15}$N heteronuclear NOE measurements of the different PRPF40A constructs (WW12truncated, WW12short, WW12, and WW12long) were performed with a 3s interscan delay; the NOE values were given by the intensity ratio of the backbone amide signals between the non-saturated and the saturated spectra. $^{15}$N $R_1$ and $R_2$ relaxation data for WW12 free and in complex with SF1$_{WWbs}$ peptide were obtained from pseudo-3D HSQC-based experiments recorded in an interleaved fashion with 10 different inversion-recovery delays (20, 60, 100, 150, 200, 300, 400, 600, 800, and 1200 ms) for $R_1$ and another 10 CPMG echo delays (16.96, 33.92, 67.84, 101.76, 135.68, 169.6, 203.52, 254.4, 339.2, and 424 ms) for $R_2$ measurement [88]. One delay was measured in duplicate for error estimation, and the relaxation rates were extracted by fitting the data to an exponential function using the relaxation module in ccpnNMR Analysis 2.5. The apparent correlation time ($\tau_c$) of WW12 free and in complex was estimated for each amide signal using the ratio of averaged R2/R1 values.

**NMR titrations.** $^{15}$N-labeled PRPF40A proteins (WW1, WW2, WW12, and N-extWW12) at around 100 μM concentration were titrated with increasing concentration ratios of polyproline or SF1 peptides (GB1 tagged) until reaching saturation, when possible; $^1$H-$^{15}$N HSQC experiments and in some cases $^1$H-$^{13}$C HSQC experiments (using double-labeled PRPF40A proteins) were acquired at each titration point. The chemical shift perturbation (CSP) was weighted using the Eqs. (1) and (2)[89].

$$CSP\,(HN) = \sqrt{0.5 \cdot \left[ \Delta\delta_H^2 + \left( 0.14 \cdot \Delta\delta_N^2 \right) \right]} \qquad (1)$$

$$CSP\,(HC) = \sqrt{0.5 \cdot \left[ \Delta\delta_H^2 + \left( 0.3 \cdot \Delta\delta_C^2 \right) \right]} \qquad (2)$$

**RDC and PRE measurements.** $^1$D$_{NH}$ RDCs were calculated by comparison of the $^1$H-$^{15}$N one bond coupling value for each residue, measured in the NMR buffer (isotropic) or in the same buffer containing 10 mg/ml Pf1 filamentous bacteriophage media (partially aligned)[90] on $^{15}$N-labeled PRPF40A WW12 samples (free and in complex with SF1$_{WWbs}$) at around 200 μM protein concentration and using doublet-separated sensitivity-enhanced $^1$H-$^{15}$N HSQC[91]. RDC data was analyzed and fitted to the PRPF40A WW12 available structures using Pales software[92].

PRPF40A WW12 mutants that incorporate cysteine residues at different positions along the sequence (E162C, S202C, and S194C) and lacking the natural cysteine (C185S) were prepared as the wild-type versions (free and bound to SF1$_{WWbs}$). Mutations did not disrupt the fold, nor the proline-rich peptide binding, as corroborated by comparison of NMR correlations for wild-type and mutant proteins (Supplementary Fig. 26). IPSL (N-(1-oxyl-2,2,5,5-tetramethyl-3-pyrrolidinyl) iodoacetamide) was added to the proteins in a 10 times molar excess in NMR buffer without DTT, incubated overnight at room temperature and protected from light. The excess of spin label was removed using a PD10 column. $^1$H-$^{15}$N HSQC experiments with a recycling delay of 3 s were acquired for the different $^{15}$N labeled PRPF40A WW12 mutants after IPSL labeling before (paramagnetic) and after (diamagnetic) the addition of a 10 molar excess of freshly prepared ascorbic acid to reduce the paramagnetic probe. Intensity ratios between paramagnetic and diamagnetic species were compared with back-calculated ratios assuming a correlation time of ($\tau_c$) 5 ns or 8 ns and a $^1$H-R$_2$ transverse relaxation rate of 50 s$^{-1}$ or 55 s$^{-1}$ for the free and bound to SF1$_{WWbs}$ states of the WW12 construct, as described before[59,93].

## Isothermal titration calorimetry (ITC)
Experiments were conducted on a MicroCal PEAQ-ITC (Malvern Instruments, UK) at 25 °C in 20 mM sodium phosphate pH 6.5 and 100 mM NaCl. PRPF40A was titrated into N-terminal GB1 tagged SF1 or polyproline constructs and vice versa with similar results. Protein concentrations in the cell were around 20–40 μM, while in the syringe ranged between 200 and 500 μM. Experiments were performed at least in duplicate with injections of 1.5 μL (0.4 μL for first point) separated by 150 s delays to recover thermal power baseline and continuous stirring in the cell (750 rpm) for correct mixing. The reference cell was filled with water in all experiments. Data were processed by removing the blank experiment (using digested GB1 fusion protein) and adjusted to one-site binding model with Malvern PEAQ-ITC Analysis software (Malvern Instruments, UK).

## Small angle X-ray scattering (SAXS)
SEC-SAXS measurements were performed at the BioSAXS beamline BM29 at the European Synchrotron Radiation Facility (ESRF) in Grenoble (France)[94], using a 2D Pilatus detector coupled with in-line HPLC. An Agilent Bio SEC-3 300A size exclusion chromatography column connected to the HPLC was previously equilibrated with a 20 mM sodium phosphate, pH 6.5, 100 mM NaCl, and 1 mM DTT buffer, samples injected at around 10 mg/ml, and the elution (0.16 ml/min) was directly exposed to the X-ray beam and the scattering detected (1 s per frame). The scattering curves for the elution peaks were merged for those points with similar calculated $R$g values and buffer subtracted.

Due to a possible effect of radiation damage causing capillary fouling in phosphate buffer runs, we also acquired the same data, in batch mode, in 20 mM HEPES, pH 6.6, 100 mM NaCl, and 1 mM DTT buffer. These measurements were performed in-house on a Rigaku BIOSAXS1000 instrument mounted to a Rigaku HF007 microfocus rotating anode with a copper target (40 kV, 30 mA). Transmissions were measured with a photodiode beam stop. Calibration was done with a silver behenate sample (Alpha Aeser). PRPF40A WW12 free and

bound to SF1$_{WWbs}$ samples were dialyzed in the HEPES buffer and measured in 3 different concentrations 0.5, 1.0, and 2.0 mg/ml in 8 frames of 900 s exposure time each. Buffer samples were measured between the protein samples and used for buffer subtraction using SAXSLab software (V3.02). The SEC-SAXS-derived and the batch-mode data are very similar (Supplementary Fig. 27). Therefore, we selected batch mode curves for further analysis; we used the 1 mg/ml (free) and 2 mg/ml (complex) as representatives due to their better Guinier analysis.

The structural parameters were analyzed using ScÅtter[95] and PRIMUS[96], and CRYSOL3 from the ATSAS suite was for the fitting of the different WW12 structures with the experimental SAXS data[97].

### Immunoprecipitation and Mass-Spectroscopy

**Plasmid production.** The PRPF40A N-terminal region (isoform 2, residues 1-221) was PCR-amplified from RPE1 cDNA (ATCC, Cat# CRL-4000) cDNA. The sequence was amplified in two fragments and both fragments were combined by overhang. All PCRs were carried out using Q5 Polymerase (NEB) according to manufacturer's protocol. The resulting PRPF40A N-terminal region was cloned into a GFP-expressing pcDNA5 vector (see pcDNA5 GFP vector in ref. [28]) using Q5 Site-Directed Mutagenesis Kit (NEB). Nter-MUT, containing the PPVP/PALPP motifs changed to AAAA/AAAAA, was obtained by mutating the N-ter-WW12-containing plasmid with Q5 site-directed mutagenesis. Plasmids were transformed into DH5alpha chemically competent cells. After culturing, DNA isolation was performed with either ZymoPURE Plasmid Miniprep Kit (Zymo Research) or QIAGEN Plasmid Plus Midi Kit (Qiagen). The sequence was confirmed by Sanger Sequencing (GATC Eurofins). All used oligonucleotides are shown in Supplementary Table 6.

**Cell Culture and transfection.** Hela cells (ATCC, Cat# CCL-2, RRID:CVCL_0030) were cultured in DMEM SILAC medium (Thermo Fisher) supplemented with 10% filtered FBS (Sigma-Aldrich) and 1% penicillin-streptomycin (Thermo-Fisher). The SILAC medium contains either light (Arg0, Lys0), medium (Arg6, Lys4) or heavy (Arg10, Lys8) isotopes (Cambridge Isotopes Laboratories). Hela cells were cultivated for 14 days in SILAC medium to assure sufficient isotope incorporation. For transfection, 7 × 10^6 Hela cells were seeded in a 15 cm culture dish in 20 ml of respective SILAC medium and grown for 24 h. The cells were then transfected with 20 µg of plasmids expressing either GFP (Control), GFP-tagged N-ext-WW12 or GFP-tagged-N-ext-WW12-MUT utilizing 64 µl FuGENE (Promega) (ratio 1:3.2), according to manufacturer's protocol.

**Immunoprecipitation.** 24 h post-transfection, cells were washed twice in ice-cold PBS and harvested in 1 ml mRIPA buffer (50 mM Tris, 150 mM NaCl, 1 mM EDTA, 1% NP-40, 0.1% Na-deoxycholate), supplemented with 2 µl Turbo-DNase (Thermo Fisher) and 3 µl protease-inhibitor cocktail cOmplete (Sigma-Aldrich). Cell lysates were cleared by centrifugation at 16,000 × g, 4 °C, 15 min and concentration was measured using Pierce BCA Protein Assay kit (Thermo Scientific). The expressed constructs were pulled down by incubating 2.5 mg of lysate with 20 µl GFP-nanobody agarose beads (ChromoTek, GTA-20) for 1 h at 4 °C under constant agitation. Subsequently, beads were washed three times with mRIPA buffer without supplements, then eluted in 1X NuPAGE LDS sample buffer (Thermo Scientific). To detach the proteins from the beads, the eluate was heated to 70 °C for 10 min To evaluate the success of the immunoprecipitation, input (1%) and IP samples were analyzed on a NuPAGE 4–12% Bis-Tris-Acrylamide gel (Thermo Fisher) using NuPAGE MOPS SDS running buffer (Thermo Fisher). Proteins were stained using Sypro Ruby protein gel stain (Thermo Fisher) according to manufacturer's protocol.

**Protein digestion.** The eluted proteins of three SILAC states (light, medium, heavy) were combined. The proteins were then reduced in DTT, alkylated in iodoacetamide, and purified using the SP3 paramagnetic bead approach[98]. This was followed by overnight enzymatic digestion using trypsin (1 µg per sample) in 50 mM ammonium bicarbonate at 37 °C. The peptide solution was then purified by solid phase extraction in C18 StageTips[99].

**Liquid chromatography tandem mass spectrometry.** Peptides were separated by online reverse phase liquid chromatography using the EASY-nLC 1000 system (Thermo Scientific) in a 30-cm analytical column (inner diameter: 75 µm; heated at 50 °C) packed in-house with ReproSil-Pur 120 C18-AQ 1.9-µm beads (Dr. Maisch GmbH). The gradient separation was performed through a 105-min non-linear gradient of 1.6–32% acetonitrile with 0.1% formic acid at a nanoflow rate of 225 nl/min. The eluted peptides were sprayed directly by electrospray ionization into a Q Exactive Plus Orbitrap mass spectrometer (Thermo Scientific). Mass spectrometry measurement was conducted in data-dependent acquisition mode using a top 10 method with one full scan (mass range: 300–1650 m/z; resolution: 70,000, target value: $3 \times 10^6$, maximum injection time: 20 ms) followed by 10 fragmentation scans via higher energy collision dissociation (HCD; normalized collision energy: 25%, resolution: 17,500 in profile mode, target value: $1 \times 10^5$, maximum injection time: 120 ms, isolation window: 1.8 m/z). Precursor ions of unassigned or +1 charge state were rejected. Additionally, precursor ions already isolated for fragmentation were excluded dynamically for 20 s.

**Mass spectrometry data processing and statistical analysis.** Mass spectrometry raw data were processed by MaxQuant software package (version 2.1.3.0)[100] using its default Andromeda search engine[101]. Mass spectra were searched against a target-decoy database consisting of the forward and reverse sequences of the bait proteins (GFP, PRPF40A_fragment_WT and PRPF40A_fragment_Mut), UniProt human reference proteome (release 2022_04; 102,601 entries) and a list of 246 common contaminants. Corresponding SILAC states were assigned (light: Arg0, Lys0; medium: Arg6, Lys4; heavy: Arg10, Lys8). Trypsin/P specificity was chosen. Carbamidomethylation of cysteine was chosen as fixed modification. Oxidation of methionine and acetylation of the protein N-terminus were set as variable modifications. A maximum of two missed cleavages were allowed. The "second peptides" option was switched on. A minimum peptide length of 7 amino acids was required. False discovery rate was set to 1% for both peptide and protein identifications.

For protein quantification, minimum ratio count was set to one. Both the unique and razor peptides were used for quantification. The "re-quantify" function was switched on. The "advanced ratio estimation" option was also chosen. For each replicate, when analyzing the M/L or H/L ratios, the log2-transformed protein ratios were normalized by adjusting to the point at which the peak density of the ratio distribution was found[102]. When analyzing H/M ratios, normalization was performed by adjusting the log2 ratios so that the bait PRPF40A_fragments would have equal abundances (H/M = 1:1).

Statistical analysis to identify differentially-regulated proteins was performed using the limma software package in R[103]. Proteins with SILAC ratios in at least two out of the three biological replicates were retained. A linear model was fitted to assess the ratios for each protein. The log2 ratio and the significance of the difference were displayed in a volcano plot. Proteins with a minimum log2 ratio of 1 and a $p$ value lower than 0.05 were considered differentially regulated.

### Reporting summary

Further information on research design is available in the Nature Portfolio Reporting Summary linked to this article.

## Data availability

NMR chemical shifts for PRPF40A tandem of WW domains free and in complex with the SF1$_{WWbs}$ peptide, as well as the backbone chemical shifts of N-terminal extended WW12 construct, are deposited in the BMRB under accession codes 34839 (WW12 apo), 34840 (WW12/SF1 complex), and 52046 (N-ext-WW12). The structure ensembles of WW12 free and bound to SF1$_{WWbs}$ are deposited in the PDB with accession codes 8PXW and 8PXX, respectively. SAXS data are deposited in the SASBDB with accession codes: SASDSK7 and SASDSL7 for the SEC-SAXS derived data of WW12 free and in complex with SF1$_{WWbs}$, respectively; SASDUH4 and SASDUJ4 for the batch mode derived curves of free and bound PRPF40A WW12. The mass spectrometry proteomics data have been deposited to the ProteomeXchange Consortium via PRIDE partner repository with the dataset identifier PXD046164. The coordinates of the truncated PRPF40A WW tandem and the WW tandem of the yeast homolog Prp40 are publicly available in the PDB under the codes 2L5F and 1O6W, respectively. ITC data, both raw curves and analysis projects, are deposited in Zenodo repository under the following https://doi.org/10.5281/zenodo.10988522. Source data are provided as a Source Data file. Source data are provided with this paper.

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

## Acknowledgements

We thank Florent Delhommel, Hyun-Seo Kang, and members of the Sattler lab for the discussions. We are grateful to Sam Asami and Gerd Gemmecker for their help with NMR experiments. We acknowledge access to NMR measurement time at the Bavarian NMR Center. We also acknowledge the European Synchrotron Radiation Facility (ESRF) for access to SAXS beamline and local contact Petra Pernot for support with SAXS data collection (bag MX2279) and Matthias Brandl for the help with in-house SAXS data acquisition. We kindly acknowledge support by the IMB Proteomics Core Facility in executing and analyzing the mass-spectrometry, as well as in-kind advice.

This work was supported by the German Research Foundation (DFG), SFB 1035, grant number 201302640 (M.S.), SPP1935 Project No. 273941853 (J.K. and M.S.) and GRK2526/1 Project No. 407023052 (J.K.). S.M.L. acknowledges a postdoctoral fellowship from the EU HORIZON 2020 research and innovation program under the MarieSkłodowska-Curie grant agreement No. 792692 (SplicEcomplex).

## Author contributions

S.M.L., L.K.T., V.D. and C.H. performed NMR experiments. S.M.L., L.K.T. and M.P. performed ITC titrations. S.M.L. performed the structure calculations and validation. M.M.M. and J.K. performed and analyzed the immunoprecipitation experiments. S.M.L. and M.S. conceived and designed the study and wrote the manuscript. All authors commented and contributed to the approved final version of the manuscript.

## Funding

## Competing interests

The authors declare no competing interests.
