## [Peer Review File · Nature Communications]

Intramolecular autoinhibition regulates the selectivity of PRPF40A tandem WW domains for proline-rich motifsREVIEWER COMMENTS

Reviewer #1 (Remarks to the Author):

To date, many aspects of multivalency in protein-protein interactions are not sufficiently understood. Regarding recognition of proline-rich peptide sequences, a multitude of not-so-specific and redundant protein sections compete for binding domains such as SH3 and WW, to mention the most prominent. Those domains occur often in multiple copies, with low differences in affinity for individual proline-rich peptides, yet they are crucial in delicate regulatory events. Given the mass of proteins exhibiting proline-rich sequences, more research is required and the authors present here a highly interesting example from the field of RNA splicing, investigating the WW-tandem in PRP40A. As a major result, the authors define a longer construct than previously investigated by Jiang et al as relevant and present a complex structure with a peptide motif from the interactions partner SF1. Specificity is investigated by ITC and NMR, including scanning of the proline-rich sections of SF1. As a surprising and equally important result, they clearly show the competition of an intramolecular proline-rich peptide that covers the binding site with the binding of SF1 sub-sequences. This corresponds to the finding that in X-ray structures of holo proteins surface-exposed proline-peptide binding sites in SH3, WW and other domains are often covered by internal sequences that may even not contain many proline residues. The authors invested considerable work in characterizing the regulatory behavior of this feature, pointing out a thresholding function. I consider this important, therefore I recommend publication in Nature Communications.

The readability needs improvement:

The abbreviation SF1 is used for many different items. Sometimes the whole protein is meant, sometimes the best-binding peptide, sometimes all sorts of peptides. I consider this tricky, since there are many binding sites in SF1. Some examples, there are many more:

- The legend of Fig. 3 says 'Structure of PRPF40A-SF1 complex'. Surely not.
- Below it says 'SF1 peptide', and the reader can try to find out which residues are visible. Ok, ok, I found out from the main text but it should be clear also from the legend.
- See headline on page 9.
- On page 10, last paragraph it says '...on SF1 binding'. Do you really mean the full protein?
- In the headline of Table S13 it says 'SF1 16 mers' and many different sequences are shown. The beginnings and ends deserve a number. To see whether the sequences are overlapping, for example. In general, in each instance of the mentioning of peptides a notation would be helpful like SF1 67-78, maybe in brackets, maybe as a subscript. Then one does not need to say SF1 peptide or SF1WWbs which gets forgotten here and there, and there are many peptides investigated. Furthermore, I did not find the exact sequences of the peptides, maybe I overlooked them: are they amidated, is there an ester at the C-terminus?

In many instances, the writing needs to be more precise. Two examples: In the text it says that signals in the center or linking region of the peptide SF1 WWbs could not be assigned since they were in slow

exchange. Please state which residues are assigned and which not, in all cases, since here and there those residues in linking regions, peptide or protein, are discussed. Along the same lines, it says on page 7: First, WW1 is slightly differently perturbed due to the interaction with either the polyproline or the PPLP motifs, confirming that the first domain interacts with the N-terminal part of the peptide. – What means slightly differently perturbed? The appearance of two signals, both shifted, after adding the peptide, as shown in Fig. 2C bottom and Fig. SI 8, bottom? Or what else? By the way, how exactly do you interpret this?

I do not buy into the new binding mode, see the sentence: These interactions involve the canonical interface of the WW domains but also reveal an unprecedented mode of recognition in this type of domain with stacking interactions of two proline residues surrounding the solvent-exposed tryptophan of WW1. – The wording describes exactly what I know about proline peptide – domain interactions, see the reviews on this matter, or describe more precisely including a detailing figure.

The discussion could cover the possibility that the binding sites of two domains could easily be on different locations of SF1 rather than in a 16-residue peptide sequence.

Furthermore, there should be more in-depth comments about the binding constants. Formally, ITC values can only be compared if the binding modes are exactly the same. However, some peptides have several possibilities for binding, along a multi-Pro sequence. I think it requires a comment.

The NMR titration experiments show interesting shifts of WW NeH. I think that those residues should be labelled in all plots since the effects are different for WW1 and 2. Also, HSQC spectra without assignment are shown. Since many residues shift it would be great to see the assignment. Yes, there are the diagrams, but still the assignment on the plot is valuable.

Since I am a seasoned person, I need to use a printout for reviewing. Light blue and small letters are no good, even worse are light grey letters (Fig. 2B). 2C bottom shows orange cross peaks, but not in the legend included. This is not intended; it occurs because of red and yellow. Better color code? Fig 6D: white letters on light green are barely visible. Fig. 4D: to some colorblind people all letters are grey. I sincerely recommend not to use light grey or blue color for letters. Black is beautiful. The authors should include the possibility that readers will use a printout.

Reviewer #2 (Remarks to the Author):

I have focused on the IP-MS elements of the manuscript as requested.

The authors performed IP-MS using GFP-tagged fusion protein of GFP-PRPF40A-Nter-WT (GFP-Nter) and GFP-PRPF40-MUT (GFP-Nter-MUT) using GFP alone as negative control. A triplex SILAC experiment was designed to establish preferential binding of proteins to either of the constructs. Samples were combined after elution of bound protein from the GFP-nanobody agarose beads. Following upstream chemistries tryptic peptides were analysed on a QExactive Plus orbitrap MS and the raw data processed

using MaxQuant with the Andromeda search engine.

Enriched proteins were identified when compared with GFP alone for both the WT and Mut PRPF40A. U2 snRNP-related proteins were highlighted as a class of protein interacting through the WW-domain of PRPF40A.

Overall these experiments were well performed and sufficient detail provided to reproduce the experiments.

There are some minor comments :

- 1) The axes on Fig 5D should be changed from Mut/WT (log₂ FC) to WT/Mut (log₂ FC)
- 2) Could the authors clarify what NO in Fig 5D refers to
- 3) It is not clear if Fig 5D is direct comparison between Mut and WT PRPF40A or is it a composite via their comparisons with GFP control.
- 4) The axes on Supp Fig 22B WT/GFP (log₂ FC) should read GFP/WT (log₂ FC)
- 5) The axes on Supp Fig 22C Mut/GFP (log₂ FC) should read GFP/Mut (log₂ FC).
- 6) Can the authors provide reciprocal Co-Ips for SF1 and both constructs.
- 7) Can the authors include in Supp Fig 22A the third biological replicate please.
- 8) Supp table 5 and Supp Fig 22 refer to Fig 5E this should be Fig 5D.
- 9) Could the authors include the number unique peptides identifying the proteins in Supp table 5. If a single peptide was identified could the authors provide annotated spectra identifying these peptides in a separate figure please.

Reviewer #3 (Remarks to the Author):

The manuscript by Martinez-Lumbreras et al. describes the interaction of the PRPF40A protein with polyproline peptides and with SF1 and SF3A1 partners. The authors solved the structure of the PRPF40A WW domains in free form and in interaction with a high affinity binding peptide by NMR. The ensemble conformations were confirmed by SEC-SAXS experiment. The interaction was also analysed by ITC and NMR using chemical shift mapping, showing that the two WW domains contribute to the interaction. Finally, they also proposed an autoinhibitory process from the N-terminal polyproline-rich region of the PRPF40A protein, which ensures a better selectivity for polyproline-rich partners.

The authors were careful to obtain enough information to contradict the previous structure of the PRPF40A WW domains. The SAXS experiments are convincing and this should be emphasised more in the summary.

The comments relate to the presentation of the data and are intended to improve the clarity of the manuscript and allow a better assessment of the quality of the SAXS samples and data:

Supplementary table 2

I strongly suggest the authors to look at the published SAXS guidelines manuscripts (10.1107/S2059798317011597 and 10.1107/S2059798322012141) with template tables for reporting BioSAXS data. The current Table 2 is not detailed enough. In this table, the method for estimating (and not calculating) the MW from the SAXS curves must be reported.

Supplementary figure 6 - C

The residuals should be added to easily compare the two fitted curves with the experimental curve. The paired distance distribution plot should be described in the legend.

Supplementary figure 11 – B

As for the Supplementary Figure 6-C, the averaged curve based on the 20 structures should be visible with its residuals.

Methods- SEC-SAXS

A reference to the ESRF beamline (10.1107/S1600577522011286) should be added.

Additional information requested.

The supplementary information should include the UV-I(0)-R_g elution profiles of the SEC-SAXS experiments. This supplemental data is essential to validate the final SAXS curves used in this manuscript.

Open question: Analysis of the binding of PRPF40A WW tandem to SF1 and SF3A1

The middle region of SF3A1 has a PPLP motif, but unlike SF1wwbs, we can observe a polyproline region preceding the PPLP motif (PPPETPMP). Is it possible that the first motif interacts with WW1 and the PPLP motif with the WW2 domain? Can you comment on this?

Point-by-point reponse to the reviewer comments

We thank the reviewers for the constructive comments and suggestions, which we have addressed in the revised version. Please find the point-by-point response below.

In addition, we have corrected typos, updated the legend to Suppl. Fig. 21 and adapted the nomenclature of SF1 constructs in the methods section to be consistent within the manuscript.

Reviewer #1

To date, many aspects of multivalency in protein-protein interactions are not sufficiently understood. Regarding recognition of proline-rich peptide sequences, a multitude of not-so-specific and redundant protein sections compete for binding domains such as SH3 and WW, to mention the most prominent. Those domains occur often in multiple copies, with low differences in affinity for individual proline-rich peptides, yet they are crucial in delicate regulatory events. Given the mass of proteins exhibiting proline-rich sequences, more research is required and the authors present here a highly interesting example from the field of RNA splicing, investigating the WW-tandem in PRP40A. As a major result, the authors define a longer construct than previously investigated by Jiang et al as relevant and present a complex structure with a peptide motif from the interactions partner SF1. Specificity is investigated by ITC and NMR, including scanning of the proline-rich sections of SF1. As a surprising and equally important result, they clearly show the competition of an intramolecular proline-rich peptide that covers the binding site with the binding of SF1 sub-sequences. This corresponds to the finding that in X-ray structures of holo proteins surface-exposed proline-peptide binding sites in SH3, WW and other domains are often covered by internal sequences that may even not contain many proline residues. The authors invested considerable work in characterizing the regulatory behavior of this feature, pointing out a thresholding function. I consider this important, therefore I recommend publication in Nature Communications.

The readability needs improvement:

The abbreviation SF1 is used for many different items. Sometimes the whole protein is meant, sometimes the best-binding peptide, sometimes all sorts of peptides. I consider this tricky, since there are many binding sites in SF1. Some examples, there are many more:

- *The legend of Fig. 3 says 'Structure of PRPF40A-SF1 complex'. Surely not.*
- *Below it says 'SF1 peptide', and the reader can try to find out which residues are visible. Ok, ok, I found out from the main text but it should be clear also from the legend.*
- *See headline on page 9.*
- *On page 10, last paragraph it says '...on SF1 binding'. Do you really mean the full protein?*
- *In the headline of Table S13 it says 'SF1 16 mers' and many different sequences are shown. The beginnings and ends deserve a number. To see whether the sequences are overlapping, for example.*

Thank you for pointing this out. We have revised the text to use clear description and nomenclature for proteins and constructs discussed. Particularly, we modified the following:

Fig.3 legend (red text is the correction):

"Figure 3. Structure of PRPF40A tandem WW domains in complex with SF1_{WWbs} peptide. (A) Cartoon representation of WW12 tandem of PRPF40A bound to the SF1_{WWbs} peptide (PPLPPGAPPPPPPPP, shown in sticks), indicating the recognition of the three segments in the peptide and the aromatic residues involved in the interaction..."

Headline on page 9:

"Analysis of the binding of PRPF40A WW tandem to SF1 and SF3A1 proline-rich regions"

Last paragraph on page 10:

"...shows the typical shifts on the WW tandem region due to SF1_{WWbs} binding, but also revealed high perturbation..."

Regarding Supplementary Table 3, peptide boundaries and nomenclature, please refer to the next comment.

In addition, we have modified the following text in line with the reviewer's suggestion:

- **Headline on page 7:**
“Solution structure of the PRPF40A tandem of WW domains in complex with SF1_{WWbs}”
- **Methods/NMR spectroscopy:**
All mentions to “SF1 peptide” have been changed to “SF1_{WWbs} peptide”.
The sentence at the end of “Protein assignment and structure calculations” subsection: has been changed to “*Final ensembles of free PRPF40A WW12 and in complex with SF1_{WWbs} were analyzed with...*”
- **Data availability:**
We have also changed “SF1 peptide” to “SF1_{WWbs} peptide”.
- **Figure 5 and its legend:**
We have added for clarity SF1_{WWbs} in panel C and in text: “**(B)** NMR titration analysis of the N-terminal extended WW12 construct with the high-affinity peptide from SF1 (SF1_{WWbs})”.
- **In Suppl. Table 2 and 4 and in legends of Suppl. Figures 9, 10, 11, 12, 13, 14 and 26:**
We have clearly labeled SF1_{WWbs} instead SF1 / SF1 peptide.

In general, in each instance of the mentioning of peptides a notation would be helpful like SF1 67-78, maybe in brackets, maybe as a subscript. Then one does not need to say SF1 peptide or SF1WWbs which gets forgotten here and there, and there are many peptides investigated. Furthermore, I did not find the exact sequences of the peptides, maybe I overlooked them: are they amidated, is there an ester at the C-terminus?

We agree with the reviewer that the notation for SF1 16-mers could be improved. Therefore, we added the beginning and end residue numbers for each peptide when the sequence is shown (Figure 2 – panel B, Figure 5 – panel C, Suppl. Table 2, Suppl. Figure 7 – panel B, Suppl. Figure 14, Suppl. Figure 20 and Suppl. Figure 21).

In the text, we now refer to the peptides with the boundaries as subscript, except for SF1_{WWbs}. We treat this peptide as special because it shows the highest affinity to SF1 and we have used it for many more experiments than the rest of peptides.

We had the sequence and boundaries information in Figures and Tables (Figure 2, Suppl. Table 2 and Suppl. Figure 7), but now we have also included in Methods.

We recombinantly produce all peptides fused to a His-GB1-TEVsite N-terminal tag; in fact, we kept the tag in the binding experiments in order to properly quantify the peptide concentrations; cleaved and purified His-GB1-TEVsite tag was used as negative control (Suppl. Figure 7). We agree with the reviewer that the chemistry of the ends could influence binding. In order to avoid effects of the C-terminal carboxyl group we always expressed the peptides with several buffer residues that are common to all peptides (polyPro and SF1 16mers): TEVsite-GAMSGS-peptide-SGSN. We have included this important information in the Methods.

In many instances, the writing needs to be more precise. Two examples: In the text it says that signals in the center or linking region of the peptide SF1 WWbs could not be assigned since they were in slow exchange. Please state which residues are assigned and which not, in all cases, since here and there those residues in linking regions, peptide or protein, are discussed.

The region of SF1_{WWbs} that could not be assigned due to signal broadening lies in the residues linking both the N and C-terminal binding sites (Pro 583 – Pro 585). We have added this information in the first paragraph of the corresponding chapter on page 7.

In fact, in the first paragraph of page 8, we briefly discuss that the structure of this region in the complex is not well-defined due to the lack of restraints as there are no visible peaks. This is the only unassigned region in the whole complex.

Along the same lines, it says on page 7: First, WW1 is slightly differently perturbed due to the interaction with either the polyproline or the PPLP motifs, confirming that the first domain interacts with the N-terminal part of the peptide. – What means slightly differently perturbed? The appearance of two signals, both shifted, after adding the peptide, as shown in Fig. 2C bottom and Fig. SI 8, bottom? Or what else? By the way, how exactly do you interpret this?

We apologize for the confusion on this aspect. This paragraph refers to the comparison of two different titrations: WW12 vs P16 (second blue bar chart in Fig2 D) and WW12 vs SF1_{WWbs} (green bar chart in Fig2 D). Both peptides are similar in the C-terminus (PPPPP) but differ at the beginning of the sequence having the SF1_{WWbs} the PPLPPGAP motif. When comparing the bound states of both titrations (1:2 final points, red bar chart in Fig 2 D), the WW2 signals overlap (there are no differences in the plot), while WW1 and the linker helix are differently perturbed (meaning that they go to different endpoints). This suggests that WW2 binds the polyproline at the C-terminus of both peptides, while WW1 binds either the polyproline in P16 or the PPLPPGAPP motif in the SF1_{WWbs} peptide.

We have modified the text of this paragraph to make it clearer and easier to read:

“When comparing the binding mode of SF1_{WWbs} (Figure 2 D, bottom, green bar chart) with the polyproline 16-mer peptide (Figure 2 D, second row, blue bar chart), differences are observed between the spectra of the bound states (Figure 2 D, third row, red bar chart). The WW2 signals overlap perfectly, suggesting that the second domain interacts with the C-terminal region of both peptides, which share a polyproline sequence. The WW1 signals are slightly differently perturbed (meaning different endpoints between both titrations) due to the interaction with either the polyproline or the PPLP motifs at the N-terminal part of the peptides. In addition, there are some chemical shift differences of the residues in the helical linker, which could be explained by the different conformations that the peptides may have in the middle region, affecting the overall arrangement of the tandem WW domains. This same region shows no perturbation in the titration with the shorter 6-mer peptide (Figure 2 D, first blue bar chart).”

The two signals in the spectra that the reviewer refers to are the side chain signals of the two Trp of WW1 and WW2 involved in binding. To avoid further confusion, we included assignment labels on the spectra, as the reviewer suggested below.

I do not buy into the new binding mode, see the sentence: These interactions involve the canonical interface of the WW domains but also reveal an unprecedented mode of recognition in this type of domain with stacking interactions of two proline residues surrounding the solvent-exposed tryptophan of WW1. – The wording describes exactly what I know about proline peptide – domain interactions, see the reviews on this matter, or describe more precisely including a detailing figure.

The sentence that the reviewer refers to is in the summary paragraph of the results section: “Solution structure of the PRPF40A tandem of WW domains in complex with SF1_{WWbs}”. We describe the common canonical recognition of the Pro-rich peptides for both domains we found in the structure of the complex (WW1 binds to PPLP and WW2 to PPPP using the canonical interface described in reviews and papers on this type of domain, see description of WW domain in last paragraph on page 4 from the introduction and the references we use there). These are shown in Fig 2 B.

However, in addition to this canonical proline recognition by single WW domains, we found more contacts established between the first WW domain and the continuation of the peptide after the PPLP motif. Pro 578 stacks with the indole ring of Trp 168 (typical for WW domains). The conformation of the peptide PGAP allows the last Proline (Pro 582) to also stack the same indole ring in the other side, sandwiching the Trp side chain (shown in Fig 3 C). This is the novelty we found in our structure. We agree that the sandwiching of a tryptophan residue by two prolines is a common recognition mode found in other structures. However, this type of interaction has not been reported for WW domain before.

We agree that the description of this interaction and the novelty in the context of WW domains has not been stated very clearly. We have therefore revised and expanded the text in the results section describing this tryptophan sandwiching interaction on page 8/9 in the revised manuscript.

We also updated the summary paragraph of the chapter:

“These interactions involve the canonical interface of the WW domains (WW1 binding to PPLP and WW2 to PPPP, respectively), but also reveal an unprecedented mode of recognition involving tryptophan sandwiching by two proline residues (in the PPGAP motif). This is enabled by the context of the tandem WW domains, where both domains interact with the extended proline-rich peptide.”

The discussion could cover the possibility that the binding sites of two domains could easily be on different locations of SF1 rather than in a 16-residue peptide sequence.

We agree with the reviewer; in fact, in the section of the results “Analysis of the binding of PRPF40A WW tandem to SF1 and SF3A1 proline-rich regions”, we explore this possibility by titrating the WW12 with larger constructs of SF1 (up to the whole C-terminal region). We do observe that the two WW domains are able to interact with two proline-rich sequences separated in the sequence with comparable affinity and stoichiometry than the corresponding 16-mer peptides:

“...the region containing the highest affinity motif SF1_{WWbs} has a comparable affinity to the isolated 16-mer, while the other regions show lower affinity, consistent with their different proline content. For example, the very C-terminal region contains two proline tracts separated by around 20 residues, which shows similar affinity (K_d around 10 mM) as the shorter 16-mers with also two complete proline tracts (Figure 2B).”

We have also included the following sentence in the paragraph in page 10:

“In summary, the WW tandem of PRPF40A recognizes a high-affinity motif in SF1 or SF3A1 but also shows some promiscuous binding to suboptimal proline-rich sequences. We have also noticed that the two WW domains are able to simultaneously interact with separated proline-rich sequences, but that the affinity is still directed by the different motifs they contain (PPLPPGAP, PPLP, PPPP). The presence of other lower affinity sequences within the same protein target could boost the interaction of both proteins through avidity effects and could preliminary load PRPF40A on SF1 until the optimal sequence is reached.”

Furthermore, there should be more in-depth comments about the binding constants. Formally, ITC values can only be compared if the binding modes are exactly the same. However, some peptides have several possibilities for binding, along a multi-Pro sequence. I think it requires a comment.

We have fitted the ITC curves and obtained the binding constants assuming a one-site binding model for all performed titrations. But as we describe in the text, we are aware that in some of the titrations, multiple register binding and more than one binding event are likely to occur. In the latter cases we observe N values different to 1 and what we obtain is an overall apparent dissociation constant. This mainly is observed with the longer peptides (PolyPro > 16 and larger regions of SF1/SF3A1). To be more precise, we refer now to “apparent K_d ” or “apparent binding affinity”, to consider that multiple events with different K_d s and in some cases with interference between binding events may occur.

The NMR titration experiments show interesting shifts of WW NeH. I think that those residues should be labelled in all plots since the effects are different for WW1 and 2. Also, HSQC spectra without assignment are shown. Since many residues shift it would be great to see the assignment. Yes, there are the diagrams, but still the assignment on the plot is valuable.

We have now added annotations to the first spectrum of Fig 2 and Fig 4.

We agree that the Trp side chains are very interesting; the ones shown in Fig. 2 and Fig. 4 correspond to the solvent-exposed Trp, which is involved in the peptide recognition; there are two other sidechain signals corresponding to the other tryptophan residue in each WW domain, which is buried (not affected by binding, in Suppl Figures). It is remarkable to see how the Trp corresponding to the first WW domain is differently affected by the Pro16 or SF1WWbs, while the one in WW2 shows no difference (related to the previous comment from the reviewer).

Since I am a seasoned person, I need to use a printout for reviewing. Light blue and small letters are no good, even worse are light grey letters (Fig. 2B). 2C bottom shows orange cross peaks, but not in the legend included. This is not intended; it occurs because of red and yellow. Better color code? Fig 6D: white letters on light green are barely visible. Fig. 4D: to some colorblind people all letters are grey. I sincerely recommend not to use light grey or blue color for letters. Black is beautiful. The authors should include the possibility that readers will use a printout.

We have modified Fig 2 and Fig 4 to include bold letters for the sequence in black. Also, prolines and SF3A1-related colors in Fig 4 are now darker. The last point of Fig 2 C titration (1:2) is indeed orange, and, as they are in slow exchange, overlap with the other colors. We have also changed the colors in Fig 6D labels.

Reviewer #2

I have focused on the IP-MS elements of the manuscript as requested.

The authors performed IP-MS using GFP-tagged fusion protein of GFP-PRPF40A-Nter-WT (GFP-Nter) and GFP-PRPF40-MUT (GFP-Nter-MUT) using GFP alone as negative control. A triplex SILAC experiment was designed to establish preferential binding of proteins to either of the constructs. Samples were combined after elution of bound protein from the GFP-nanobody agarose beads. Following upstream chemistries tryptic peptides were analysed on a QExactive Plus orbitrap MS and the raw data processed using MaxQuant with the Andromeda search engine.

Enriched proteins were identified when compared with GFP alone for both the WT and Mut PRPF40A. U2 snRNP-related proteins were highlighted as a class of protein interacting through the WW-domain of PRPF40A.

Overall these experiments were well performed and sufficient detail provided to reproduce the experiments.

There are some minor comments :

We thank the reviewer for the comments for improving the IP-MS results. Please find our detailed responses to each point below:

1) The axes on Fig 5D should be changed from Mut/WT (log2 FC) to WT/Mut (log2 FC)

We revisited the axis description and verified their correctness. Because we compared the mutated PRPF40A to its WT version, the proteins significantly enriched in PRPF40A-Nter-MUT appear on the positive side of the scale, and therefore the right side of the figure. Nevertheless, in order to increase the readability of the figure, we improved the figure legends. The axis now reads: "PRPF40A-Nter-MUT vs. PRPF40A-Nter-WT".

2) Could the authors clarify what NO in Fig 5D refers to

In the legend of Figure 5D we state that the dots in gray are non-significant binding differences. To avoid misunderstandings and clarify the annotation of the figure, we changed Figure 5 D from "NO" to "Not significantly enriched".

3) It is not clear if Fig 5D is direct comparison between Mut and WT PRPF40A or is it a composite via their comparisons with GFP control.

We apologise that this was unclear. Figure 5D is a composite analysis including enrichment over the GFP control. We have changed the legend to make this clearer:

"(D) Plot comparing interactors of GFP-tagged PRPF40A-Nter-WT vs PRPF40A-Nter-MUT construct. In the PRPF40A-Nter-MUT construct, the proline-rich sequence was substituted by a polyalanine stretch. To remove background, only proteins that are significantly enriched for the GFP-tagged WT- or Mut-PRPF40A constructs compared to GFP alone are considered. Significantly altered protein interactions are

shown in blue ($\log_2 FC > \log_2(1.5)$, adjusted p -value < 0.05), non-significant binding differences are shown in grey. Adjusted p -values were determined by linear modeling using the limma package.” Additionally, we explain the detailed procedure in the methods section.

4) The axes on Supp Fig 22B WT/GFP ($\log_2 FC$) should read GFP/WT ($\log_2 FC$)

Please see our explanation to comment 1.

5) The axes on Supp Fig 22C Mut/GFP ($\log_2 FC$) should read GFP/Mut ($\log_2 FC$).

Please see our explanation in comment 1. For clarity, labels of Suppl Fig 22B & C have been modified as in Fig 5D

6) Can the authors provide reciprocal Co-IPs for SF1 and both constructs.

The intention of the IP-MS data was not to prove the interaction between the N-terminal PRPF40A region and SF1, which has been shown in previous works. Instead, we wanted to specifically understand what effect a, in principle, non-disrupting mutation in the intrinsically disordered region of PRPF40A could have on the PRPF40A interaction network. We believe that performing the reciprocal Co-IPs using SF1 as bait would provide little information for the autoinhibitory mechanism of PRPF40A that we describe in the manuscript.

7) Can the authors include in Supp Fig 22A the third biological replicate please.

We apologise that this was unclear. Actually, we performed the pulldown experiments in 5 replicates. Two replicates were used for the SDS-PAGE analysis and three were analyzed via MS. Therefore, we cannot provide further replicates of the same experiment. For clarification, we renamed the replicates and address them as “replicate 4” and “replicate 5”.

8) Supp table 5 and Supp Fig 22 refer to Fig 5E this should be Fig 5D.

We thank the reviewer for spotting this error. We have corrected this.

9) Could the authors include the number unique peptides identifying the proteins in Supp table 5. If a single peptide was identified could the authors provide annotated spectra identifying these peptides in a separate figure please.

We added the number of unique peptides for each protein in Supp Table 5 in the column “Unique peptides”. One of the proteins (FAM120A) was identified by only a single identifying peptide. The spectrum has been added as Supp Figure 22D. All other proteins were identified by at least 2 peptides.

Reviewer #3 (Remarks to the Author):

The manuscript by Martinez-Lumbreras et al. describes the interaction of the PRPF40A protein with polyproline peptides and with SF1 and SF3A1 partners. The authors solved the structure of the PRPF40A WW domains in free form and in interaction with a high affinity binding peptide by NMR. The ensemble conformations were confirmed by SEC-SAXS experiment. The interaction was also analysed by ITC and NMR using chemical shift mapping, showing that the two WW domains contribute to the interaction. Finally, they also proposed an autoinhibitory process from the N-terminal polyproline-rich region of the PRPF40A protein, which ensures a better selectivity for polyproline-rich partners.

The authors were careful to obtain enough information to contradict the previous structure of the PRPF40A WW domains. The SAXS experiments are convincing and this should be emphasised more in the summary.

We thank the reviewer for appreciating our effort to show that the optimized construct, which comprises all structured regions around the tandem domains, clearly displays no interdomain interactions. We propose to modify the abstract in the following way:

“Here, we combine NMR, SAXS and ITC to determine the structure of the PRPF40A tandem WW domains in solution and characterize the binding specificity and mechanism for proline-rich motifs recognition.”

Related to this topic, we have adapted the following sentence in the first paragraph of the discussion (page 13) as we cannot be certain about whether the differences in the Jiang et al structure relate to using a truncated protein or incorrect NOE assignments:

“This structure is differs from a previously reported structure³⁸, in which a truncation of the C-terminal helix or erroneous automated NOE assignments may have impaired the structural determination.”

The comments relate to the presentation of the data and are intended to improve the clarity of the manuscript and allow a better assessment of the quality of the SAXS samples and data:

Supplementary table 2

I strongly suggest the authors to look at the published SAXS guidelines manuscripts (10.1107/S2059798317011597 and 10.1107/S2059798322012141) with template tables for reporting BioSAXS data. The current Table 2 is not detailed enough. In this table, the method for estimating (and not calculating) the MW from the SAXS curves must be reported.

We have updated the table following the guidelines.

Supplementary figure 6 – C

The residuals should be added to easily compare the two fitted curves with the experimental curve. The paired distance distribution plot should be described in the legend.

We have modified Suppl Fig 6 to include the residuals and the SEC profile; also we have modified the corresponding legend.

Supplementary figure 11 – B

As for the Supplementary Figure 6-C, the averaged curve based on the 20 structures should be visible with its residuals.

We have modified Suppl Fig 11:

- We have added the SEC profiles together with the Rg values and the selected frames.
- We have replaced the 20 curves by the averaged one.
- We have included the residuals for the SAXS fitting.
- Legend was updated.

Methods- SEC-SAXS

A reference to the ESRF beamline (10.1107/S1600577522011286) should be added.

The reference has been added.

Additional information requested.

The supplementary information should include the UV-I(0)-Rg elution profiles of the SEC-SAXS experiments. This supplemental data is essential to validate the final SAXS curves used in this manuscript.

The SEC profiles with the Rg values for each peak are shown now in the respective supplementary figure. The merged frames were also display for clarity.

Open question: Analysis of the binding of PRPF40A WW tandem to SF1 and SF3A1
The middle region of SF3A1 has a PPLP motif, but unlike SF1_{wwbs}, we can observe a polyproline region preceding the PPLP motif (PPPETPMP). Is it possible that the first motif interacts with WW1 and the PPLP motif with the WW2 domain? Can you comment on this?

The binding modes could be multiple, and not necessary within 16 residues (the longer the distance between the two binding sites for both WW domains, the lower the cooperativity effect). In the case of this peptide from SF3A1, there are 3 putative binding sites for a single WW domain: the PPPET region at the N-terminus, the PPPLPP motif in the middle and a PLPPAP sequence far downstream in the sequence. WW1 could bind to the first and WW2 to the second (or third) as the reviewer points, and also WW1 could bind to the second (PPLP) and the WW2 to the C-terminal one. Our aim was to point that in the case of SF3A1 there is no higher apparent affinity compared to SF1, probably because of the absence of the extended PPLP motif (PPLPPGAP).

We have updated this part of the manuscript to briefly (due to length limitations) discuss this observation by adding the following sentence (in page 10):

“We performed ITC titrations of this SF3A1 region (364-413) with the WW tandem of PRPF40A, obtaining comparable binding in the low micromolar range, although with lower apparent affinity than SF1_{wwbs} ($K_d \approx 10 \mu\text{M}$). This peptide contains 3 main proline-rich regions, being the PPLP motif the middle one; complicating the binding analysis and interpretation as multiple binding events may occur. In any case, the absence of the PPLPPGAP sequence may be an explanation of the lower affinity this SF3A1 region shows compared to the SF1 constructs containing the SF1_{wwbs} sequence.”

REVIEWER COMMENTS

Reviewer #1 (Remarks to the Author):

The authors responded well to all requests for improvements. I recommend publication without delay.

Reviewer #2 (Remarks to the Author):

The authors have addressed adequately all the concerns raised in the review.

Reviewer #3 (Remarks to the Author):

Thank you to the authors for addressing my comments and providing the requested information. Upon reviewing the new data, I have some reservations regarding the quality of the obtained data:

Comment 1:

In supplementary table 2, for both constructions, the minimum measured q -range is equal to 0.01 nm^{-1} , while the q_{min} used for the Guinier estimation is approximately 0.03 nm^{-1} . Could you please explain the removal of the initial points from the curve? It would also be desirable to provide the figure with the Guinier fit and the residuals.

Comment 2:

Figure S6 (C) and figure 11 (B): Regarding the SEC-SAXS profiles, the buffer signal at the end of elution is higher than at the beginning of elution. This is likely due to radiation damage causing capillary fouling during elution. This fouling over time can lead to the generation of SAXS curves not solely corresponding to the sample signal and generally accompanied by an increase in signal at very small angles. The use of a phosphate buffer may explain this drift. I recommend the authors to redo the measurements in another buffer without phosphate to validate the results.

Comment 3:

Some units are in nm^{-1} and others are in \AA^{-1} in the tables and figures. For clarity, it would be preferable to choose only one.

Typo corrections:

Table S2

Sample injection volume instead of Sample injection volumen

Method for scaling intensities instead of Method for scaling **intesities**

Point-by-point reponse to the reviewer comments

We thank reviewer #3 for the careful review and suggestions. We have followed the advice and performed in house SAXS measurement in HEPES buffer, which are fully consistent with the previous data and conclusions.

We have adapted the manuscript to show the new data in HEPES buffer (Fig. 1 and Suppl. Fig. 6 and 11), while also keeping the synchrotron data and comparing these with the new data in a new Supp. Figure 27.

Reviewer #3

Thank you to the authors for addressing my comments and providing the requested information. Upon reviewing the new data, I have some reservations regarding the quality of the obtained data:

Comment 1:

In supplementary table 2, for both constructions, the minimum measured q-range is equal to 0.01 nm⁻¹, while the q_{min} used for the Guinier estimation is approximately 0.03 nm⁻¹. Could you please explain the removal of the initial points from the curve? It would also be desirable to provide the figure with the Guinier fit and the residuals.

The averaged curves were noisy in the initial points (low q values) probably due to the problem with buffer subtraction (see below) and with no obvious trend (no aggregation or interparticle repulsion). We therefore decided to show the measurements in HEPES buffer, which are very comparable and do not change overall conclusions. In addition, we provide the Guinier fit and residuals for those curves in the corresponding Supp. Fig. 6 and 11. Note, that upon uploading the data to SASDB, the automated quality check removes the first points in the SEC-SAXS curves as well.

Comment 2:

Figure S6 (C) and figure 11 (B): Regarding the SEC-SAXS profiles, the buffer signal at the end of elution is higher than at the beginning of elution. This is likely due to radiation damage causing capillary fouling during elution. This fouling over time can lead to the generation of SAXS curves not solely corresponding to the sample signal and generally accompanied by an increase in signal at very small angles.

The use of a phosphate buffer may explain this drift. I recommend the authors to redo the measurements in another buffer without phosphate to validate the results.

We agree with the comment that the use of phosphate buffer may have caused the noise we observe at small angles.

Following the suggestion, we have repeated the measurement of both PRPF40A WW12 free and in complex with SF1_{WWbs} in similar buffer but using HEPES instead phosphate. We have updated the manuscript accordingly. We decided to keep the SEC-SAXS measurements in the new Supp. Fig. 27 as the curves fully overlap.

With the new data in HEPES buffer, the structural validation of both the free and the bound states of PRPF40A and the conclusions drawn remain unchanged.

Comment 3:

Some units are in nm⁻¹ and others are in Å⁻¹ in the tables and figures. For clarity, it would be

preferable to choose only one.

Typo corrections:

Table S2

Sample injection volume instead of Sample injection volumen

Method for scaling intensities instead of Method for scaling intesities

Thanks for the suggestions, all units are in Å.

REVIEWERS' COMMENTS

Reviewer #3 (Remarks to the Author):

The authors have effectively addressed all improvement requests. I recommend proceeding with publication promptly.